# A FOUNDATION MODEL FOR WEATHER AND CLIMATE

## ABSTRACT

Triggered by the realization that AI emulators can rival the performance of traditional numerical weather prediction models running on HPC systems, there is now an increasing number of large AI models that address use cases such as forecasting, downscaling, or nowcasting. While the parallel developments in the AI literature focus on foundation models – models that can be effectively tuned to address multiple, different use cases – the developments on the weather and climate side largely focus on single-use cases with particular emphasis on mid-range forecasting. We close this gap by introducing Prithvi WxC, a 2.3 billion parameter foundation model developed using 160 variables from the Modern-Era Retrospective Analysis for Research and Applications, Version 2 (MERRA-2). Prithvi WxC employs an encoder-decoder-based architecture, incorporating concepts from various recent transformer models to effectively capture both regional and global dependencies in the input data. The model has been designed to accommodate large token counts to model weather phenomena in different topologies at fine resolutions. Furthermore, it is trained with a mixed objective that combines the paradigms of masked reconstruction with forecasting. We test the model on a set of challenging downstream tasks namely: Autoregressive rollout forecasting, downscaling, gravity wave flux parameterization, and extreme events estimation.

## 1 INTRODUCTION

Deep learning is increasingly transforming weather applications by delivering highly accurate forecasts with reduced computational costs compared to traditional numerical weather prediction methods (Bi et al., 2023; Lam et al., 2023; Mukkavilli et al., 2023). Unlike the traditional physics-based approaches, deep learning models do not directly simulate the underlying physics. Instead, they capture this through probability distributions derived from model training, a method adapted from natural language processing and computer vision. This technique has proven surprisingly effective in approximating complex physical systems such as the weather. However, most current deep learning models for weather are *task-specific* forecast emulators, which focus solely on the forecasting problem. (See, however, Koldunov et al. (2024).) Key examples include FourCastNet (Pathak et al., 2022), Pangu-Weather (Bi et al., 2022), GraphCast (Lam et al., 2022), FengWu (Chen et al., 2023), Stormer (Nguyen et al., 2023b) and AIFS (Lang et al., 2024). Machine learning models also show promise for longer-term subseasonal-to-seasonal forecasts (Weyn et al., 2021). Additionally, ML-based approaches are being explored to enhance climate predictions (see Mansfield et al., 2023; Eyring et al., 2024, for a review), with a focus on the development of ML-driven parameterizations (Rasp et al., 2018; Zhao et al., 2019; Espinosa et al., 2022; Yuval & O'Gorman, 2023; Henn et al., 2024; Gupta et al., 2024), bias corrections (Bretherton et al., 2022; Gregory et al., 2024), and assessments of climate change impacts (Davenport & Diffenbaugh, 2021; Diffenbaugh & Barnes, 2023, among others). There is fascinating emerging work that combines the strengths of the data-driven and physics-based approaches (Kochkov et al., 2024; Husain et al., 2024; Roy et al., 2024). Finally, there are further large, task-specific models for nowcasting (Andrychowicz et al., 2023) and downscaling (Mardani et al., 2024).

Looking beyond atmospheric sciences at developments in AI in general and language models in particular, the last few years have been dominated by the emergence of foundation models. That is, large AI models pretrained in a task-agnostic manner that can be effectively fine-tuned to address a number of specific use cases. Despite the mirroring successes of large AI models in both fields, applications of the foundation model principle to atmospheric sciences have been rare. ClimaX (Nguyen et al., 2023a) and AtmoRep (Lessig et al., 2023) considered problems ranging from

nowcasting to downscaling and bias corrections; Aurora (Bodnar et al., 2024) focusses a number of different *forecasting* problems.

To address this gap, we introduce Prithvi WxC, a large-scale foundation model for weather and climate applications trained on the Modern-Era Retrospective analysis for Research and Applications, Version 2 (MERRA-2) (Gelaro et al., 2017). Prithvi WxC is a transformer-based deep learning architecture which combines ideas from several recent transformer architectures in order to effectively process regional and global dependencies of the input data and to capture longer sequence lengths of tokens. Moreover, the model is capable of running in different spatial contexts. In addition we introduce a new pretraining objective that blends masking and forecasting.

The validation of Prithvi WxC extends from zero shot evaluations for reconstruction and forecasting to downstream tasks such as downscaling of weather and climate models, the prediction of hurricane tracks and atmospheric gravity wave flux parameterization.

## 2 PRITHVI WXC

Prithvi WxC has been designed to address several questions that arise when considering the meaning of foundation models for atmospheric physics: Since weather models can run on the entire earth or in a regional context, do we need specialized architectures for global and local problems? Do we need to differentiate between models with zero and non-zero lead time? If we do consider tasks with zero and non-zero lead time, what is a suitable pretext task for pretraining?

### 2.1 PRETRAINING OBJECTIVE

*Forecast emulators* are typically trained by predicting the state of the atmosphere $X_{t+\delta t}$ at time $t + \delta t$ given the state at times $t$ and $t - \delta t$. Some do so by directly regressing on physical quantities. For most the output is the tendency $X_{t+\delta t} - X_t$. Foundation models for vision on the other hand are frequently based on masked autoencoders (He et al., 2022). This is notable since masking is a natural pretext task for weather and climate as observational data is ungridded and sparse. Indeed, note that the emerging literature on models working directly on observation makes heavy use of masking (Vandal et al., 2024; McNally et al., 2024). Moreover, the forecasting task breaks down for $\delta t = 0$ while foundation model use cases satisfy no such constraint.

In the end our pretraining objective combines masking with forecasting. Moreover, while our lack of constraints on $\delta t$ makes it impossible to predict tendencies, we generalize the objective of Lam et al. (2022) to another source of free information, namely Climatology. Instead of predicting the difference from the current time stamp $X_t$, we model the deviation from historical climate at this time, $C_t$. All in all, our pretraining objective is

$$\frac{\hat{X}_{t+\delta t} - C_{t+\delta t}}{\sigma_C} = f_\theta \left[ M_{0.5} \left( \frac{X_t - \mu}{\sigma}, \frac{X_{t-\delta\tau} - \mu}{\sigma} \right); \frac{C_{t+\delta t} - \mu}{\sigma}, S, \delta t, \delta\tau \right]. \quad (1)$$

Here, $f_\theta$ is the model, $\hat{X}_{t+\delta t}$ the prediction at time $t + \delta t$, $\mu$ and $\sigma$ are per variable means and standard deviations (computed across space and time). $\sigma_C^2 = \sigma_C^2(X_t - C_t)$ is the variance of the historical anomaly; again computed across space and time. $S$ are static inputs and $\delta t$ and $\delta\tau$ are the time steps for the target and the inputs respectively. Finally, $M_{0.5}$ denotes the masking operator.

### 2.2 DATA

We pretrain our model on 160 variables from the MERRA-2 reanalysis. Here, we use 3-hourly data from 20 surface variables as well as 10 variables from 14 model native levels. In addition, we use four static variables such as surface geopotential height and land fraction. See A.1.1 for a complete list of variables and levels. We train the model using data from 1980 to 2019. We validate with data from one of the years in the 2020-2023 range, depending on task.

The climatology appearing in equation 1 was computed using data from 2000 to 2019. We follow the methodology of (Janoušek, 2011) except that our climatology resolves the diurnal cycle. (See appendix A.1.2.)

Regarding normalizations, we found that leaving $\sigma$ and $\sigma_C$ in equation 1 unconstrained leads to instabilities. This is essentially due to the large range of values we have, especially the anomalies in the mass fraction of cloud liquid water QL at high model levels can be as small as $10^{-26}$. In light of this we impose $10^{-4} \leq \sigma \leq 10^4$ and $10^{-7} \leq \sigma_C \leq 10^7$. This mainly affects $Q_I$ and $Q_L$ at high levels.

## 2.3 ARCHITECTURE

At it's core, Prithvi WxC is a 2D vision transformer. To keep it as flexible as possible, we aimed not to use architecture elements that restrict to "rectangular" topologies for data. (Even though we train on MERRA-2 data on a rectangular lat/lon grid, one can envision training or running inference directly on Gaussian grids.) Vanilla ViTs would satisfy this requirement, yet do not scale to large token counts. Considering the different flavors of scalable transformers we notice the findings of "Hiera" (Ryali et al., 2023). Here the authors show that it is possible to surpass the performance of Swin transformers (Liu et al., 2021) with a more flexible and simplified architecture. Turning back to AI models for weather, Andrychowicz et al. (2023) made use of MaxViT (Tu et al., 2022) leverages axial attention. In the end, our core idea is that if we pretrain the model using only attention, we keep the core of the model flexible and can add convolutions at fine-tuning time to increase performance when suitable. We do so by joining the approaches of Hiera and MaxViT.

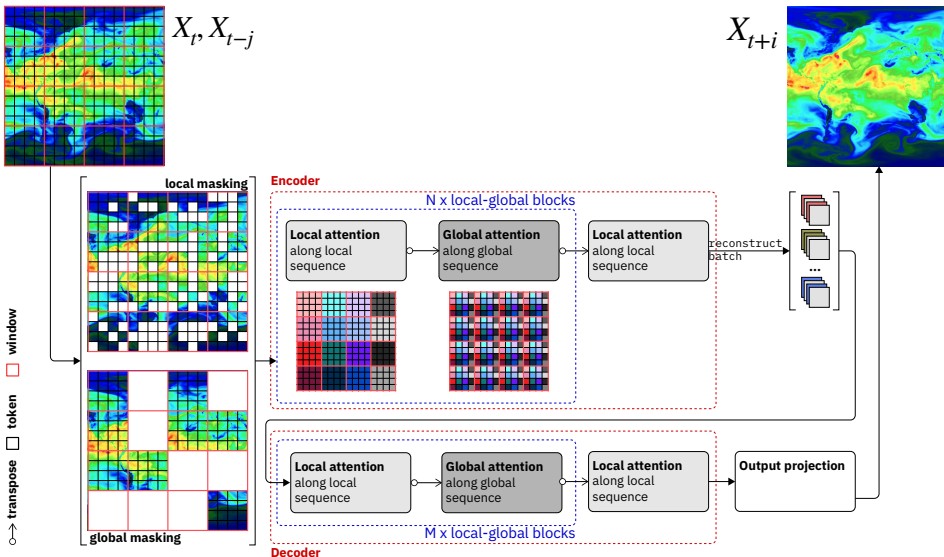

Figure 1: Prithvi WxC core architecture elements and masking scheme. For simplicity the figure ignores elements such as embedding and output layers as well as position encodings.

In detail, the only constraint we impose on the data is the ability to structure tokens into windows – akin to Swin, Hiera and MaxViT. After tokenization, our data takes the shape (batch, windows, tokens, features), where the second dimension enumerates windows and the third tokens within each window. Subsequently we alternate attention within a window and across windows. Modulo masking, the latter is similar to (Tu et al., 2022). In what follows we will refer to attention within a window as "local" and attention across windows as "global". When masking, we can either mask out entire global windows or individual tokens within a window. A byproduct of the latter is that global attention no longer connects the same token in each window. For an illustration of the attention and masking pattern see figure 1.

As shown in equation 1, the model has several inputs: To start, there are the *dynamic* inputs $X_t$, $X_{t-\delta\tau}$. These take the shape

$$T \times [V_S + (V_V \times L)] \times H \times W = 2 \times [20 + (10 \times 14)] \times 360 \times 576 = 320 \times 360 \times 576.$$

$T$, $L$, $H$ and $W$ denote time, vertical level, latitude and longitude respectively. $V_S$ and $V_L$ enumerate surface and model level parameters. $C_{t+\delta\tau}$ take the shape $160 \times 360 \times 576$ as there is the same

number of parameters yet no time dimension. The static inputs $S$ are based on 4 static parameters from MERRA-2 as well as cosine and sine of day of year and hour of the day. We use a static Fourier position encoding that respects the periodicity of the earth. In addition, there is a learned encoding for both the lead time $\delta t$ as well as the input time step $\delta\tau$.

In its final configuration Prithvi WxC comprises 25 encoder and 5 decoder blocks. The internal dimension is 2,560. This results in 2.3 billion parameters. With a token size of 2 by 2 pixel we are dealing with 51,840 tokens per sample. For full details regarding model configuration and memory consumption see appendix B.3.

## 2.4 PRETRAINING

We train Prithvi WxC in two phases. The first uses a 50% masking ratio and alternates "local" and "global" masking. Moreover, we randomize the time steps for targets and inputs ($\delta t$ and $\delta\tau$) choosing among 0, 6, 12 and 24 as well as 3, 6, 9 and 12 respectively. This phase uses 64 A100 GPUs and results in a highly flexible model which we use for our downscaling and gravity wave parametrization experiments as well as for the zero-shot reconstruction evaluations.

The second phase makes a few optimizations to attune the model for forecasting applications: We reduce the masking ratio to 50%, fix $\delta t = \delta\tau = 6$ and introduce a Swin-shift (Liu et al., 2021) in the now dense encoder. Following Lam et al. (2022) we introduce weights to training objective equation 1, yet weigh u10m and v10m higher than t2m. See appendix B.4 for details. This version of the model is used for the forecast evaluation as well as the hurricane-forecasting use case.

## 2.5 ZERO-SHOT VALIDATION

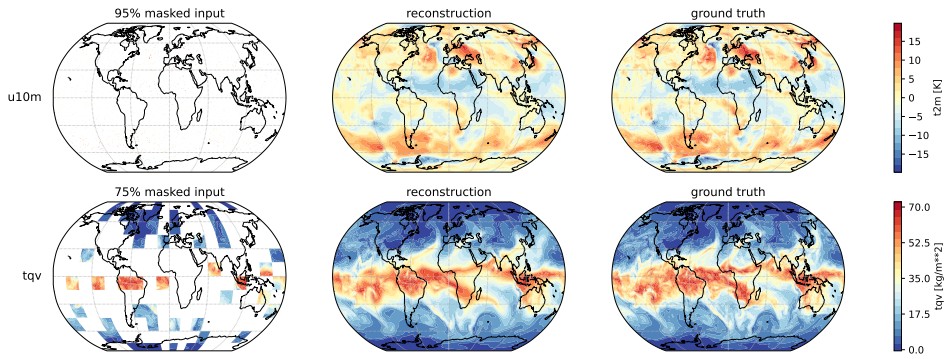

Figure 2: Zero-shot reconstruction performance with Prithvi WxC. The first row shows "local" masking where we mask 95% of individual tokens. The second row shows "global" masking where we mask 75% of attention windows.

**Reconstruction** We start the zero-shot evaluation with reconstruction task. Figure 2 shows examples of reconstruction from locally and globally masked data. Note that the model is capable of reconstructing atmospheric state from as little as 5% of the original data when the samples are still relatively dense and 25% when we mask out large areas. Corresponding RMSE scores can be found in figure 3. It is interesting that reconstruction performance is relatively little affected by lead time at the lower end of masking ratios. This opens up the possibility of initializing a forecast model with randomly sampled tokens to obtain an ensemble forecast as well as the future research direction to fine-tune the model to integrate sparse observational data.

**Forecasting** Next we perform autoregressive forecasts with dense data up to 5 days ahead. See figure 12. To put our results into context, we compare data from various AI forecast emulators as well as the ECMWF IFS as provided by WeatherBench2 (Rasp et al., 2024). In addition, we compare with a version of FourCastNet (Pathak et al., 2022) trained on MERRA-2 data. The validation period is 2020. Some care has to be taken when interpreting these results. WeatherBench2 compares against ERA5 and the IFS Analysis at 0.25 degrees resolution while we work with MERRA-2 at 0.5 by

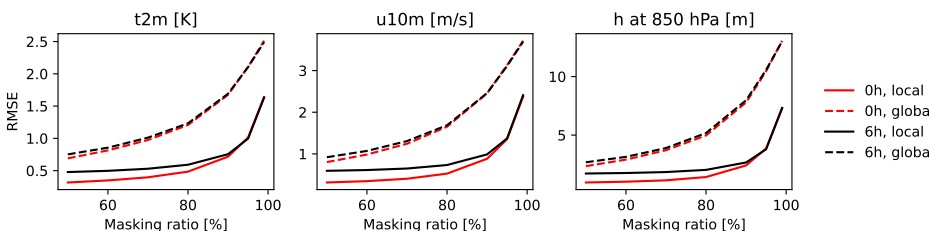

Figure 3: Zero-shot reconstruction performance of Prithvi WxC.

0.625. Moreover, our model generates a number of forecasts for which no reference AI prediction exists. Most notably the "cloud" variables. With all these caveats in mind, Prithvi WxC performs well to exceptionally well at very short lead times (6 and 12 hours), particularly for parameters like surface temperature. However, performance then decays and after about 66 hours Prithvi WxC falls below the performance of Pangu-Weather.

The reader might remark that we should not refer to this as zero shot performance when the model has gone through rollout tuning. However, we expect that one should do several things when truly pushing for maximal forecasting performance. Among these are adding additional convolutional or neural operator layers that improve information flow from attention window to attention window as well as deeper rollout tuning.

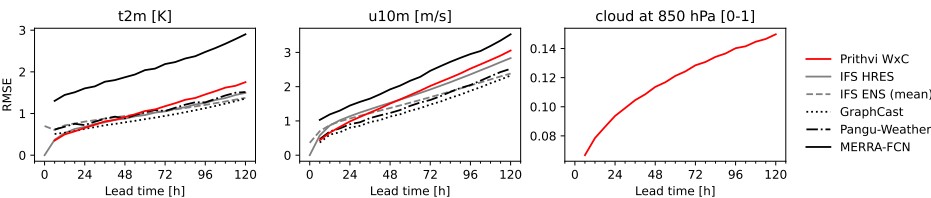

Figure 4: Zero-shot forecasting performance of Prithvi WxC.

**Hurricane track forecasting**   We validate Prithvi WxC to assess its capability in forecasting the formation, dissipation, intensification, and tracking of hurricanes ranging from Category 3 to Category 5, formed over the Atlantic Ocean between 2017 and 2023. The list of hurricanes used in the analysis is provided in Table 10. In this task we benchmark against observed hurricane tracks from the HURDAT database and two other models: FourCastNet trained on MERRA-2, and FourCastNet trained on ERA5.

Figure 5 gives an example, figure 6 a comprehensive assessment over a five-day forecast for all the hurricanes included in table 10. By the end of the five-day forecast, the Prithvi WxC's track error is 200 km less than that of the benchmark models. While Prithvi WxC outperforms the MERRA-2 trained FourCastNet in MSLP and windspeed predictions, it is marginally outperformed by the ERA5 trained FourCastNet, likely due to the finer spatial resolution in the ERA5 dataset.

## 3   Prithvi WxC: Downstream Validation

In what follows we will look at a number of downstream applications realized via fine-tuning. In all examples the pretrained part of the model remains frozen. We will see changes of the dataset, changes in spatial and temporal resolution, changes in selected variables and pressure levels and finally a change of spatial domain. Given this variability we always add new embedding and output layers and sometimes select other architecture elements.

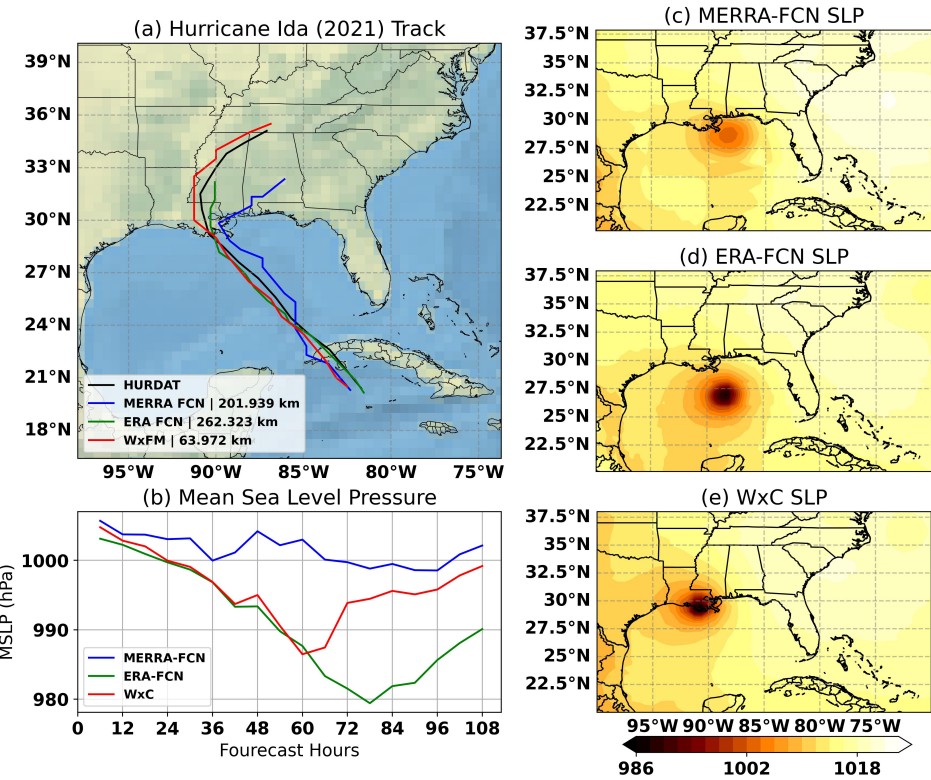

Figure 5: Hurricane Ida (2021). (a) Hurricane track. All models were initialized at 00 UTC on 2021-08-27. (b) 5-day forecast of Mean Sea Level Pressure (MSLP). (c-e) Sea Level Pressure (SLP) for a 60-hour forecast (12 UTC on 2021-08-29).

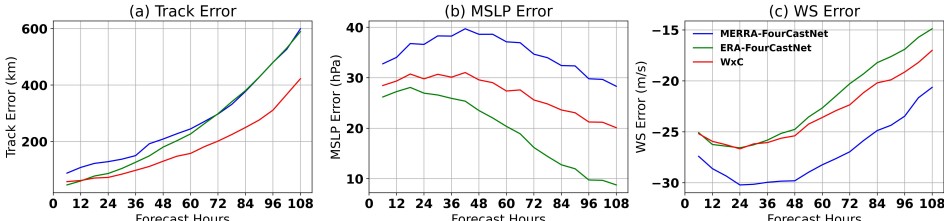

Figure 6: 5 days composite (75 difference initial conditions) forecast of track errors, MSLP errors and WS errors from MERRA-FourCastNet, ERA-FourCastNet and Prithvi WxC models

## 3.1 DOWNSCALING

**MERRA-2**   We fine-tune Prithvi WxC to increase the spatial resolution of weather and climate model data. When doing so, we use the architecture of figure 13. That is, we embed the pretrained transformer in a series of convolution and pixel shuffle layers. Thus, we can increase resolution both before and after the pretrained (and frozen) transformer. To validate the overall downscaling performance in a clean setup that isolates model performance from dataset questions, we finetune a 6x weather downscaling model for 2m surface temperature using MERRA-2 data. The input data variables are the same as used for pre-training. We first coarsen MERRA-2 data from dimension 361 x 576 (50km x 62.5km resolution) to dimension 60 x 96 (300km x 375km resolution), and secondly apply a smoothing operation in form of a convolution with a 3x3 pixels kernel.

Figure 7 visualizes the downscaling performance for a single timestamp. Following the Climate-Learn benchmark (Nguyen et al., 2024) we compare the model performance with interpolation base-

lines. Here the model performance is evaluated on the entire validation period between 2021-01-01 and 2021-12-30 and results are summarized in Table 1. Compared to the interpolation baselines, Prithvi WxC improves spatial and temporal RMSE values by over a factor of 4 and also shows the best temporal correlation. In comparison, the ClimateLearn benchmark reports improvements by over a factor of 2 when downscaling 2x coarsened ERA5 2m temperature.

Table 1: Performance evaluation of the Prithvi WxC downscaling model. Spatial RMSE, temporal RMSE, and temporal correlation is evaluated on MERRA-2 2m air temperature (t2m) for a 1-year period from 2021-01-01 to 2021-30-12 (2,912 samples); and on CORDEX near-surface air temperature (tas) for a 95-years period from 2006-01-01 to 2100-12-31 (34,688 samples) on the RCP4.5 scenario.

| | MERRA2 - t2m (K) | | | CORDEX - tas (K) | | |
|---|---|---|---|---|---|---|
| | sp. RMSE | tp. RMSE | tp. corr. | sp. RMSE | tp. RMSE | tp. corr. |
| Nearest | 3.22 | 2.46 | 0.89 | 2.07 | 1.46 | 0.99 |
| Bilinear | 3.08 | 2.34 | 0.90 | 2.01 | 1.41 | 0.99 |
| Prithvi WxC | **0.73** | **0.64** | **0.98** | **0.49** | **0.44** | **1.00** |

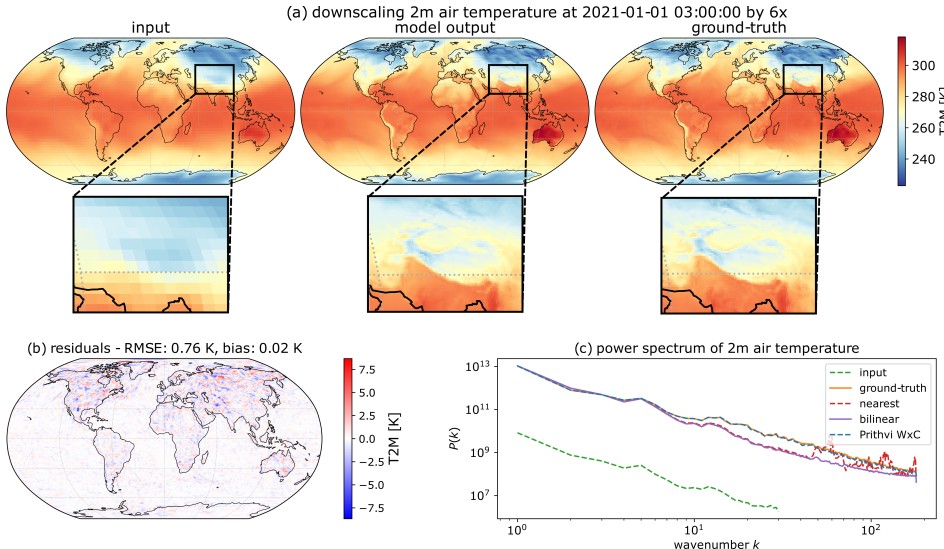

Figure 7: Downscaling MERRA-2 2 meter air temperature for 2021-01-01 at 3 UTC.

**CORDEX** We now switch from a global to a regional context as we focus on data from the Coordinated Regional Climate Downscaling Experiment (CORDEX). Specifically, we use a subset of data from the EURO-CORDEX simulations (Jacob et al., 2014) at a resolution of 0.11° x 0.11° (12.5 km x 12.5 km) covering a domain over Europe. In contrast to the case of previous section 3.1, this changes the dataset, the temporal step as well as the domain from the pretraining case. We fine-tune a 12x climate downscaling model for daily mean near-surface air temperature for a period from 2006 to 2100 under scenario RCP8.5 (Moss et al., 2010). Input variables are shown in Table 5. We coarsen the input data of dimension 444 x 444 (12.5 km x 12.5 km resolution) to dimension 37 x 37 (150 km x 150 km resolution) and apply a smoothing convolution as previously.

Model performance is evaluated on data from simulation scenario RCP4.5 which was not seen during training. Results of a single timestamp are shown in Figure 8. We evaluated the model performance over the entire simulation period from 2006-01-01 to 2100-12-31. The average metrics displayed in Table 1 indicate improvements over baseline interpolation methods of spatial and temporal RMSE values by factors of around 4 and 3, respectively. Temporal correlation values are generally high across all methods which is most likely explained by the fact that downscaling is done on daily mean values of near-surface air temperature. In their *perfect model world* experiment, Doury et al. (2023) report a mean spatial RMSE of 0.55 K. Our mean spatial RMSE is 0.49 K. When comparing

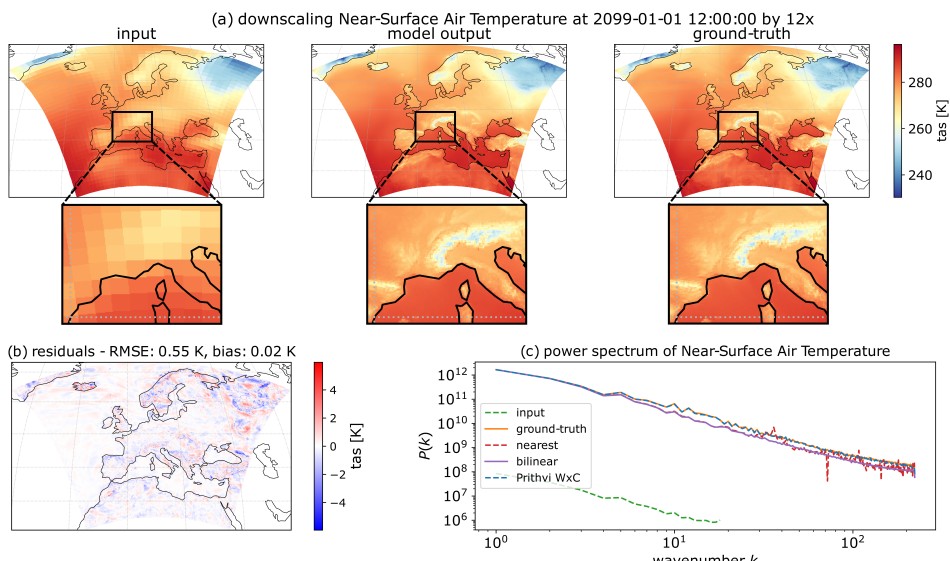

Figure 8: Downscaling CORDEX near-surface air temperature (tas) for 2099-01-01 at 12 UTC.

the two, note that our work does not include the 1-D covariates resulting from solar, ozone and anthropogenic greenhouse gas forcings. Moreover, the results presented here are calculated on a bigger spatial domain (corresponding approximately to the EURO-CORDEX simulation domain) without masking the sea.

## 3.2 CLIMATE MODEL PARAMETERIZATION FOR GRAVITY WAVE FLUX

Atmospheric gravity waves (GWs) are intermittent, small-scale ($\mathcal{O}(1)$ to $\mathcal{O}(1000)$ km) perturbations generated around thunderstorms, jet disturbances, flows over mountains, etc. They belong to a class of physical processes crucial to the earth's momentum budget but only crudely represented in coarse-climate models which rely on inadequate *physical parameterizations* instead of resolving them. As such, the fine-tuning task is to use the latent space of Prithvi WxC to develop data-driven physical parameterizations to provide missing sub-grid scale variability in coarse-climate models at zero-lag. For this task, the model is fine-tuned using high-fidelity, high-resolution gravity wave data extracted from ERA5 (which resolves a substantial portion of the atmospheric gravity waves).

We use four years of ERA5 global reanalysis on the 122 lowest vertical model levels and 30 km horizontal resolution at hourly-frequency to prepare the training data for fine-tuning. The model takes the zonal wind speed ($u$), meridional wind speed ($v$), temperature ($T$), and pressure ($p$), along with positional variables latitude, longitude, and surface height as input. The model outputs the directional momentum fluxes carried by gravity waves mathematically expressed as the covariances ($u'\omega', v'\omega'$), and are computed using Helmholtz decomposition using the horizontal $(u,v)$=(U,V) and vertical wind speeds ($\omega$=OMEGA). Both the input and output are conservatively coarse-grained to a $64{\times}128$ ($\approx 300$ km) latitude-longitude grid to be consistent with a typical coarse-climate model and to remove phase dependencies of the calculated fluxes.

The architecture schematic for the fine-tuning is a U-net like architecture shown in Figure 14: The frozen encoder is preceded by 4 learnable convolution blocks each with an increasing number of hidden channels. Likewise, the frozen decoder is succeeded by 4 new learnable convolution blocks. Since gravity wave flux prediction is an instantaneous flux calculation task, we fix the lead time $\delta t$ to zero. The model input now has shape [1, 488, 64, 128] where the 488 channels comprise the four background variables $u$, $v$, $t$ and $p$ on 122 vertical levels each, and on a $64 \times 128$ horizontal grid, as discussed above. The model was fine-tuned to produce an output with shape [1, 366, 64, 128] comprising of the potential temperature, $u'\omega'$, and $v'\omega'$ on 122 vertical levels each. We emphasize that Prithvi WxC was pretrained on MERRA-2 data but is now fine-tuned with ERA5 data.

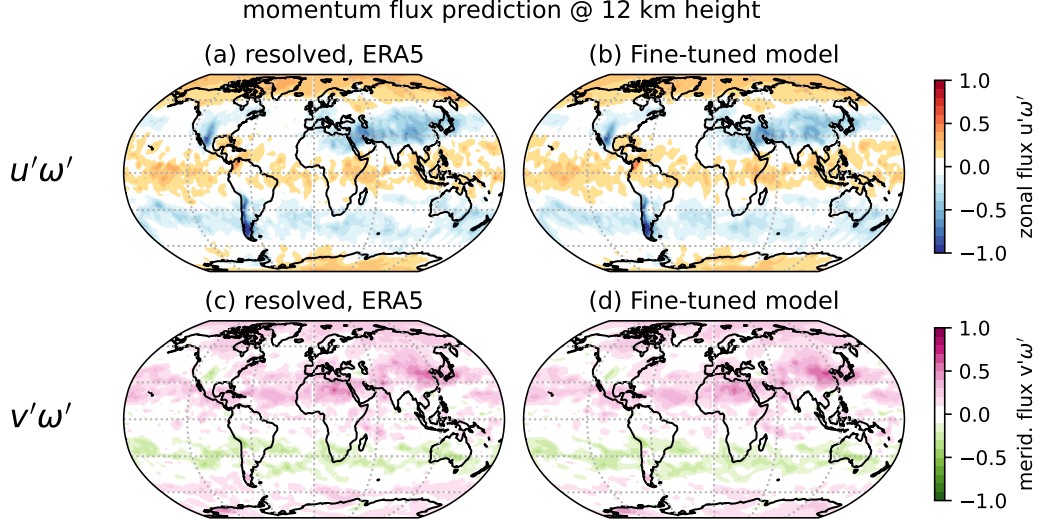

Figure 9: True vs. predicted (non-dimensionalized) momentum fluxes in the upper troposphere (12 km height) for the gravity wave flux parameterization downstream task. All fluxes are monthly averaged for May 2015. The vertical derivative of the fluxes represents the wind-forcing tendencies due to gravity waves in the atmosphere and can be used to represent a portion of unresolved sub-grid tendencies in climate models.

As a straightforward test, we look at the climatological distribution, i.e., the monthly-averaged momentum fluxes in the upper troposphere, and compare the spatial distribution of the predicted directional fluxes with the validation data from ERA5 (Figure 9). The prediction from the model closely agrees with the true flux distribution in the upper troposphere. The nature and properties of the waves over land can be significantly different from waves over the ocean. Therefore, getting a strong agreement over both the ocean and the land indicates effective learning. For instance, enhanced fluxes over the Rocky Mountains, the Andes, and the Himalayas indicates the fine-tuned model skillfully predicts the stationary waves generated over mountain ranges. Likewise, the tropical band of positive flux (in Figure 9b) in the tropics points to effective learning of non-stationary gravity waves generated around intense convective and precipitation systems. In fact, the fine-tuning model outperforms task-specific baselines created using MLPs and Attention U-Nets.

Without loss of generality, the same finetuning procedure can be applied to develop parameterizations for other sub-grid atmospheric processes of relevance to climate; albeit with some tweaks. A coarse climate model with a typical resolution of $\mathcal{O}(100)$ km fails to capture most gravity wave effects (or clouds, or fine-scale turbulence) due to its inability to resolve the smaller-scales. Owing to periodic data assimilation and higher-resolution, numerical weather prediction models are largely unaffected by these biases. Running climate models at a high-resolution over multiple centuries, however, is computationally not so feasible. To address this, we have proposed one climate-focused application of Prithvi WxC and demonstrated its effectiveness. This model can subsequently be integrated with coarse-resolution climate models of varying complexity to account for the "missing" gravity wave physics and correct the physics tendencies. The accuracy of the predicted fluxes also points to the remarkable effectiveness of the fine-tuning process in blending task-specific data from heterogenous sources.

## 4  CONCLUSIONS

This study introduces Prithvi WxC, a 2.3 billion parameter foundation model designed for weather and climate applications. Trained on 160 atmospheric variables from the MERRA-2 dataset, Prithvi WxC leverages a scalable and flexible transformer-based architecture to capture both regional and global dependencies in atmospheric data. Prithvi WxC addresses a diverse set of downstream tasks,

aligning with the foundation model paradigm prevalent in AI research. To achieve this, the model introduces a new architecture and novel objective function. The latter combines masked reconstruction with forecasting, incorporating climatological information to enhance its generalizability.

The zero-shot evaluation introduces reconstruction as a new benchmark and reveals that the model excels in forecasting at shorter lead times. We hypothesize that this strength stems from the masking objective, which encourages Prithvi WxC to grasp atmospheric dynamics with limited temporal progression.

When it comes to fine-tuning, it is important to highlight the diversity of datasets, parameters, and resolutions addressed in the downscaling and parameterization examples. In both cases, we demonstrate that a pretrained, frozen transformer trained on a single dataset can be effectively combined with additional architectural components to achieve strong results on new tasks with different datasets. Furthermore, the CORDEX downscaling case showcases the model's ability to operate in both global and regional contexts, a characteristic that we attribute to the heavy use of "global" masking during pretraining.

Even though there is no previous work on AI-based downscaling using MERRA-2, we chose this example to isolate the model's and architecture's downscaling performance from questions of distribution shift when changing datasets. Here, we found that the fine-tuned Prithvi WxC model improves by more than a factor of 4 over interpolation baselines. This 6x downscaling compares to an improvement factor of 2 when doing 2x downscaling with ERA5 data in the ClimateLearn benchmarks. That is, we have doubled the performance for a threefold resolution increase, evidence of strong performance. This is mirrored by the more applicable CORDEX example which compares favorably to the results of Doury et al. (2023).

Finetuning Prithvi WxC also demonstrates that large transformer-based foundation models can effectively learn mesoscale atmospheric evolution, helping to streamline, enhance, and accelerate the development of physical parameterizations in climate models, which in turn improves prediction accuracy on interannual timescales. The fine-tuned model produces strong predictions across all six hotspots, including both the relatively smoother fluxes over the Andes, Southern Ocean, New-foundland, and the Scandinavian Mountains, as well as the more turbulent fluxes over the Pacific Ocean and Southeast Asia. Notably, for the Andes (mountain waves) and the Southern Ocean (non-mountain waves), the fine-tuned model achieves correlation coefficients of 0.99 and 0.97, respectively, when compared to the observed fluxes.

The latent encoder-decoder space of Prithvi WxC foundation model captures a comprehensive understanding of atmospheric evolution by training on vast amounts of data, including winds, temperature, humidity, radiation, and soil moisture. Instead of building task-specific ML-models from scratch, these pretrained encoders can be used to develop more precise data-driven models of atmospheric processes.

## REPRODUCIBILITY STATEMENT

The code for is included in the supplementary materials. Model checkpoints will be made available via Hugging Face.

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

# A  DATA

## A.1  DATA

### A.1.1  MERRA-2

The Modern-Era Retrospective Analysis for Research and Applications Version 2 (MERRA-2) (Gelaro et al., 2017), developed by NASA's Global Modeling and Assimilation Office (GMAO), serves as the primary dataset for this study. It uses a cubed-sphere grid, which results in uniform grid spacing at all latitudes. This design minimizes grid spacing irregularities found in latitude-longitude grids, enhancing the dataset's spatial consistency and usefulness for global-scale analyses. MERRA-2 provides a comprehensive and consistent record of Earth's climate and atmospheric conditions, offering valuable insights into long-term climate trends and variability. It is a state-of-the-art reanalysis dataset that integrates a range of observational data with advanced modeling techniques to produce a high-quality, multidecadal record of atmospheric conditions (Rienecker et al., 2011; Gelaro et al., 2017). It is particularly useful for climate research due to its extensive historical coverage and sophisticated data assimilation methods.

Table 2: List of Surface Variables

| Variable | Collection | Description |
|----------|------------|-------------|
| u10 | M2I1NXASM | 10 m zonal wind |
| v10 | M2I1NXASM | 10 m meridional wind |
| t2m | M2I1NXASM | 2 m surface temperature |
| qv2m | M2I1NXASM | 2 m specific humidity |
| ps | M2I1NXASM | Surface Pressure |
| slp | M2I1NXASM | Sea Level Pressure |
| ts | M2I1NXASM | Skin Temperature |
| tqi | M2I1NXASM | Column-total ice |
| tql | M2I1NXASM | Column-total liquid water |
| tqv | M2I1NXASM | Column-total watre vapor |
| gwetroot | M2T1NXLND | Rootzone soil wetness relative to soil holding capacity |
| lai | M2T1NXLND | Leaf area index |
| eflux | M2T1NXFLX | Surface latent heat flux |
| hflux | M2T1NXFLX | Surface sensible heat flux |
| z0m | M2T1NXFLX | Surface roughness |
| lwgem | M2T1NXRAD | Longwave radiation emitted by the surface |
| lwgab | M2T1NXRAD | Longwave radiation absorbed by the surface |
| lwtup | M2T1NXRAD | Upward longwave at the top of atmosphere |
| swgnt | M2T1NXRAD | Net downward shortwave radiation at the surface |
| swtnt | M2T1NXRAD | Net shortwave at top of atmosphere |

Our pretraining dataset includes variables at model native levels corresponding to nominal pressure surfaces which are 985 hPa, 970 hPa, 925 hPa, 850 hPa, 700 hPa, 600 hPa, 525 hPa, 412 hPa, 288

Table 3: List of Native Vertical Level Variables

| Variable | Collection | Description |
|---|---|---|
| u | M2I3NVASM | Wind speed/direction |
| v | M2I3NVASM | Wind speed/direction |
| omega | M2I3NVASM | Vertical motions |
| t | M2I3NVASM | Air temperature |
| qv | M2I3NVASM | Specific humidity |
| pl | M2I3NVASM | Actual mid-level pressure |
| h | M2I3NVASM | Mid-layer height (equivalent to the geopotential height) |
| cloud | M2I3NVASM | Cloud fraction at this layer for radiation |
| qi | M2I3NVASM | Cloud mass fraction that is ice |
| ql | M2I3NVASM | Cloud mass fraction that is water |
| **Nominal Pressure (hPa)** | | |
| 985 \| 970 \| 925 \| 850 \| 700 \| 600 \| 525 \| 412 \| 288 \| 245 \| 208 \| 150 \| 109 \| 48 | | |

Table 4: List of Static Variables

| Variable | Dataset | Description |
|---|---|---|
| phis | M2C0NXASM | Surface geopotential height |
| frland | M2C0NXASM | Fraction of surface that is land |
| frocean | M2CONXCTM | Fraction of surface that is ocean |
| fraci | M2CONXCTM | Fraction of surface that is ice |

hPa, 245 hPa, 208 hPa, 150 hPa, 109 hPa, and 48 hPa, with data available every 3 hours. Variables at these levels include wind components (u, v), vertical wind ($\omega$), air temperature (t), specific humidity (qv), actual mid-level pressure (pl), and mid-layer geopotential height (h), cloud fraction (cloud), cloud massk fraction that is ice (qi) and water (ql).

Additional single-level variables are available at 1-hour intervals and include near-surface wind components (u10, v10), near-surface (2 meter) air temperature (t2m), skin temperature (ts), surface roughness (z0m), specific humidity (qv2m), surface pressure (ps), sea level pressure (slp), column-total ice, liquid water and water vapor (tqi, tql, tqv), longwave radiation emitted by the surface (lwgem), longwave radiation absorbed by the surface (lwgab), upward longwave at the top of atmosphere (lwtup), net downward shortwave radiation at the surface (swgnt) and net shortwave at top of atmosphere (swtnt). Static variables include surface geopotential height (phis), land fraction (frland), ocean fraction (frocean), and ice fraction (fraci), which are used to provide essential static information, and is varying in space, but not time. Time-averaged variables, such as rootzone soil wetness (gwetroot), leaf area index (lai), and surface fluxes (eflux, hflux), are aggregated from 1-hourly intervals, because these are the diagnostics variables and not available at the analysis time. Aggregation methods are used for variables from hourly products, where means of adjacent hourly values are used to create 12:00 UTC data. For example, the mean of 11:30 and 12:30 values is calculated to prepare the 12:00 UTC data. Missing values (NaNs) in gwetroot and lai are replaced with 1 and 0, respectively, to maintain data availability over the ocean. Static datasets are incorporated by creating monthly files, ensuring that the static variables (phis, frland, frocean, fraci) remain consistent for each month, thereby maintaining the integrity of static information throughout the dataset. List of variables used in the training is listed in the tables 4, 3 and 2. We train the model using data from 1980 to 2019. We validate with data from one of the years in the 2020-2023 range, depending on task.

### A.1.2 CLIMATOLOGY

The climatology appearing in equation 1 was computed from 20 years of MERRA-2 data following the methodology of the ERA-Interim climatology (Janoušek, 2011). That is, for each Julian day and each hour of the day we aggregate all data across the last 20 years. Subsequently we apply a 61-day rolling window weighted average to this. The weights are given by a second order polynomial. Thus

the climatology resolves the day-night cycle. There are $365 \times 8$ timestamps and each pixel is based on $20 \times 61 = 1220$ data points. We used the same 20 year period that we used for training; that is 1980-2019.

### A.1.3 NORMALIZATION

While equation 1 is a fairly natural training objective, we found that leaving the normalization constants $\sigma$ and $\sigma_C$ unconstrained leads to instabilities during training. This is essentially due to the large range of values we have, especially the anomalies in the mass fraction of cloud liquid water QL at high model levels can be as small as $10^{-26}$ at level 34. To avoid such extreme values upsetting numerics, we impose $10^{-4} \leq \sigma \leq 10^4$ and similarly $10^{-7} \leq \sigma_C \leq 10^7$. In both cases, this mainly affects $Q_I$ and $Q_L$ at high levels.

### A.1.4 CORDEX

For this particular downscaling experiment, we use a subset of data from the EURO-CORDEX simulations (Jacob et al., 2014) at a resolution of $0.11°$ x $0.11°$ (12.5 km x 12.5 km) covering a domain over Europe (EUR-11 CORDEX) and based on the regional climate model CNRM-ALADIN63 (Nabat et al., 2020), which is driven by the global climate model CNRM-CM5 (Voldoire et al., 2013). A list of input variables can be found in table 5.

Table 5: List of CORDEX variables. Experiments use daily mean values of scenario simulations RCP4.5 and RCP8.5 between 2006 and 2100.

| Variable | Level (hPa) | Unit | Description |
|---|---|---|---|
| hus500, hus700, hus850 | 500, 700, 850 | - | Specific Humidity |
| ta500, ta700, ta850 | 500, 700, 850 | K | Air Temperature |
| ua500, ua700, ua850 | 500, 700, 850 | m/s | Eastward Wind |
| va500, va700, va850 | 500, 700, 850 | m/s | Northward Wind |
| zg500, zg700, zg850 | 500, 700, 850 | m | Geopotential Height |
| psl | surface | Pa | Sea Level Pressure |
| tas | surface | K | Near-Surface Air Temperature |
| uas | surface | m/s | Eastward Near-Surface Wind |
| vas | surface | m/s | Northward Near-Surface Wind |

## B ARCHITECTURE AND PRETRAINING

### B.1 ARCHITECTURE

The architecture is capable of using non-geospatial tokens as context tokens. Indeed, an earlier version of the model used such a context token for the lead time. By visualizing attention patterns it became clear that this led to the emergence of specialized transformer layers that paid heavy attention to this token, which is in conflict with stochastic depth (drop path) which we enabled during the scaling phase. Thus we replaced the lead time token with a lead time embedding (and equivalent for the input time delta).

Finally there are separate linear embedding layers for the dynamic inputs as well as the concatenation of the climatological and static ones. Once embedded, all tokens are added up.

### B.2 ABLATIONS STUDIES AND ARCHITECTURE CHOICES

#### B.2.1 PRETRAINING OBJECTIVE

**Climatology** A key differentiator in our pretraining objective equation 1 is the explicit use of climatology. As an ablation study, we train two preliminary yet identical versions of the model for 24h on 16 GPUs. Naturally, we cannot compare the loss yet we can compare RMSE values for physical quantities. Since equation 1 is a mixed training objective combining forecasting with masking, let us state for completeness that this ablation involves 50% masking as well as lead time

of $(-24, -12, -6, -3, 0, +3, +6, +12, +24)$ hours. (The final version of our model used only non-negative lead times.) Table 6 shows the impact for a variety of variables as well as lead times. There is clear, often significant improvement across the entire set. One might not be surprised that highly volatile 10 meter wind speed profits the least from this.

Table 6: Use of climatology (anomalies) in the pretraining objective

| Variable | Lead time | Absolutes | Anomalies | Improvement |
|---|---|---|---|---|
| t2m [K] | 0h | 1.21 | 0.77 | 36% |
| | 6h | 1.30 | 1.30 | 24% |
| u10m [m/s] | 0h | 0.84 | 0.89 | 6% |
| | 6h | 1.16 | 1.10 | 5% |
| h at 850 hPa [m] | 0h | 8.65 | 14.52 | 40% |
| | 6h | 15.09 | 10.01 | 34% |

We did notice that one can improve beyond climatology by predicting tendencies – as explained in the main text and as was done in e.g. (Lam et al., 2022). We chose not to do this since this pretext task breaks down for the case of $\delta t = 0$. Naturally one could speculate whether a model pretrained with strictly positive lead times can be tuned to zero lead time. Moreover, it is curious that (Bodnar et al., 2024) made use of the "absolute" pretraining objective (i.e. making use of neither tendencies nor climatology) with great results. Thus, it is perceivable that the advantages of both tendencies and climatology diminish as one trains these models further.

**The effect of masking**    The other key differentiator of the pretraining objective is the use of masking. The main body of the text motivated this via the conceptual similarity to the data assimilation problem where data is sparse, memory efficiencies as well as its widespread adoption in computer vision. Table 7 shows the results of detailed ablation studies.

Before discussing the results, let us give a concise statement of the four different experiments compared here. To start, we have Prithvi WxC as well as the rollout tuned version of Prithvi WxC, which are discussed in the main body of this paper. In addition, we have two version which are architecturally identical to Prithvi WxC that have been trained on 16 GPUs for 24 hours using 0 and 50% mixed global/local masking respectively. This is the same masking procedure using in Prithvi WxC. This means that training used lead times of 0, 6, 12 and 24 hours ahead as well as deltas between the input time stamps of 3, 6, 9 and 12 hours.

To validate, we compute the average reconstruction error under 50% global or local masking across 0 and 6 ahead. Furthermore, we compute forecasting performance for either 6 hours ahead or averaged across 6, 12 and 24 hours ahead.

Table 7: Impact of masking

| | Local rec. | Global rec. | 6 h ahead | 6, 12, 24 h ahead |
|---|---|---|---|---|
| 0% masking | 0.220 | 0.251 | 0.072 | 0.102 |
| 50% masking | 0.060 | 0.105 | 0.082 | 0.110 |
| Impact of masking | 73% | 58% | $-14\%$ | $-8\%$ |
| Prithvi WxC | 0.028 | 0.068 | 0.042 | 0.070 |
| Prithvi WxC Rollout | 0.040 | 0.109 | 0.036 | 0.090 |
| Impact of rollout | $-43\%$ | $-60\%$ | 14% | $-29\%$ |

With this in mind, the ablation studies and evaluations show that masking effectively trades off a small amount of forecasting performance ($-8$ and $-14\%$) for a significant gain in reconstruction capability (58 and 73%). Table 7 also explains our rationale to publish both the pretrained as well as the rollout-tuned model. While rollout tuning improves 6-hour ahead forecasting performance by a significant 14%, local and global reconstruction capability as well as the combined 6, 12 and

24-hour ahead forecasting performance drop significantly. One might want to compare the latter to the approach of (Nguyen et al., 2023b).

### B.2.2 ARCHITECTURE

**Handling of lead time signal** Our model has both an encoder and decoder component. The two are architecturally identical except that data in the encoder is masked. Once is encoded, mask tokens are introduced for the much shallower decoder where data is now dense. This follows the paradigm of (He et al., 2022) exactly.

Given the objective of equation 1, one can wonder whether one should introduce the lead time signal at the encoder stage or only at the decoder stage. The rationale for doing so would be for the encoder to learn a representation of the data independent of lead time. Once again we trained on 16 GPUs for 24 hours. Table 8 shows the result. By introducing the lead time already in the encoder pretraining loss improves by 3%. However, one has to be careful with this result: In principle we are interested in performance for *downstream* tasks. So it is perceivable that a more difficult pretext task where lead time information is only introduced in the decoder leads to more powerful representations in the encoder. Still, given the result of this particular ablation, we chose to introduce a lead time signal already in the encoder. Incidentally one should note the results of (Nguyen et al., 2023b) which found optimal results when using adaptive layer normalization to handle varying lead times. This follows similar findings in other domains. See e.g. (Peebles & Xie, 2023).

Table 8: Handling of lead time signal

| Decoder | Encoder | Improvement |
| --- | --- | --- |
| 0.110 | 0.107 | 3.0% |

**Use of context tokens** Our architecture allows for easy integration of explicit context tokens. Indeed, an earlier version of our model (trained without drop path) handled the lead time signal discussed in the previous section using an explicit context token. Figure 10 shows attention patterns between the geospatial tokens as well as said context token in both the encoder and decoder. What is striking is the emergence of specialized layers paying high attention to said token. This is in line with findings by (Touvron et al., 2021). Here, the authors showed that the presence of class tokens in ViT layers leads to a "conflict of interest" for the layers as the task of informing the class token is different from the task of understanding the data. Figure 10 shows highly similar behavior. Layers 4, 5 and 8 pay a lot of attention to the lead time token, yet the others not. In order to avoid the emergence of such specialized layers we drop the lead time token and introduce lead time via a simple Fourier encoding.

**Scaling** Among the key advantages of transformer architectures are their scaling behaviors. In particular, (Kaplan et al., 2020) made a detailed study of scaling laws for language models, showing clear power law behavior when scalig compute, data and model size. A proper analysis of scaling behavior of Prithvi WxC is beyond the scope of this paper as a correct comparison requires training to convergence. To somehow capture this, we compare two versions of Prithvi WxC trained on 16 GPUs for an identical number of gradient descent steps – namely $12,400$. One with 2.3B parameters, the other with 280M. In both cases we keep the number of heads as well as the MLP multiplier constant at $4$ and $16$ respectively. Similarly, both are trained with $5\%$ drop path. As shown in 9, we find $12\%$ improvement in pretraining loss when increasing the model size by a factor of about $8$ by modifying the encoder depth and embedding dimension. Compare this to the findings of (Bodnar et al., 2024) which shows an improvement of about $5\%$ with every doubling of parameters. For pushing transformer architectures to extreme scales see also (Wang et al., 2024).

Regarding the number of blocks in table 9, note that we always start end end the encoder (decoder) with a local attention block. Thus, having $N$ global blocks leads to $2N+1$ total transformer blocks. In the table this is captured via the $N(2N+1)$ notation.

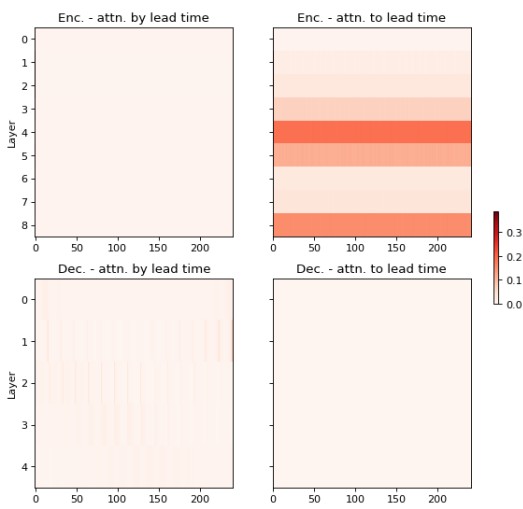

Figure 10: Attention patterns between spatial tokens and non-spatial lead-time token.

Table 9: Scaling behavior of Prithvi WxC.

| Parameters [M] | Encoder blocks | Decoder blocks | Embedding dim. | Loss |
|---|---|---|---|---|
| 280 | 8 (17) | 2 (5) | 1,024 | 0.1109 |
| 2,300 | 12 (25) | 2 (5) | 2,560 | 0.0972 |

## B.3 PRETRAINING

In its pretraining configuration Prithvi WxC comprises 25 encoder and 5 decoder blocks. As both the encoder and decoder start and end with local attention, 13 (3) of these blocks perform local and 12 (2) global attention respectively. The internal dimension is 2,560. With 16 attention heads and an MLP multiplier of 4 this results in 2.3 billion parameters. We use a token size of 2 by 2 pixel. Each window measures 30 by 32 pixel or 15 by 16 tokens. With these choices we are dealing with 51,840 tokens per sample yet are keeping the length of the global and local sequence roughly balanced. Note that both token and window size can be changed when tuning the model and we will do so repeatedly below. The model now consumes a bit more than 43 GB of GPU memory in pretraining. If we keep masking at 50% we are able to backpropagate through 4 autoregressive steps on a 80 GB A100. (Masking only applies to the first autoregressive step.) If the data becomes dense (i.e. 0% masking) this reduces to 3 steps. Since the data becomes dense in the decoder and our pretraining data does live on a rectangular grid, we add a Swin-shift to the decoder layers. The overall scale was chosen to ensure that autoregressive "rollout" training is still possible.

We make use of Fully Sharded Data Parallelism (FSDP) as well as flash attention (via scaled dot product attention). We train the model with bfloat16 precision. However, to ensure numeric stability we only use bfloat16 for the transformer layers. The input and output layers remain at float32. Finally, we use activation checkpointing. For validation-time inference we tested both float32 and bfloat16. While float32 comes with a considerable speed penalty, we observed no signifidant accuracy gains.

## B.4 PRETRAINING PROTOCOL

We train Prithvi WxC in two phases. The first phase uses 5% drop path, a 50% masking ratio and alternates "local" and "global" masking from gradient descent step to gradient descent step. Moreover, for each sample we select a random forecast lead time (among 0, 6, 12 and 24 hours ahead) as well as a random delta between inputs (-3, -6, -9, -12). With this randomization, we train the model on 64 A100 GPUs and batch size 1 for 100,000 gradient descent steps. After 2,500 steps of linear warm-up we perform cosine-annealing from $10^{-4}$ to $10^{-5}$. This results in a highly flexible

model that we use for our downscaling and gravity wave parametrization experiments as well as for the zero-shot reconstruction evaluations.

To further attune the model to forecasting applications, we make a few changes: We reduce the masking ratio to 0% and add a Swin-shift to the encoder. Also, we set drop path to 0%. In addition, we fix both the forecast lead time and input delta to six hours so that there is no more randomization. Keeping the learning rate constant at $10^{-5}$, we tune the model with 1, 2 and 3 autoregressive steps on a varying compute footprint ranging from 16 to 48 GPUs. In this phase we also modify the training objective equation 1 by using additional weights. For the vertical parameters, weights depend linearly on pressure level (in hPa). In addition, we weight H, $\omega$, T, U and V with 1 yet cloud, PL, QI, QL with 0.1. For the surface parameters, we weigh u10m and v10m with 1, SLP and t2m with 0.1 and the remaining parameters with 0.01. Essentially this follows (Lam et al., 2022) with the exception that we found it beneficial to swap the weights for t2m and u10m as well as v10m while suppressing all variables which are not standard in the AI-forecast emulation literature by a factor of ten. This version of the model is used for the forecast evaluation as well as the hurricane-forecasting use case.

## C    ZERO-SHOT VALIDATION

### C.1    RECONSTRUCTION AND FORECASTING

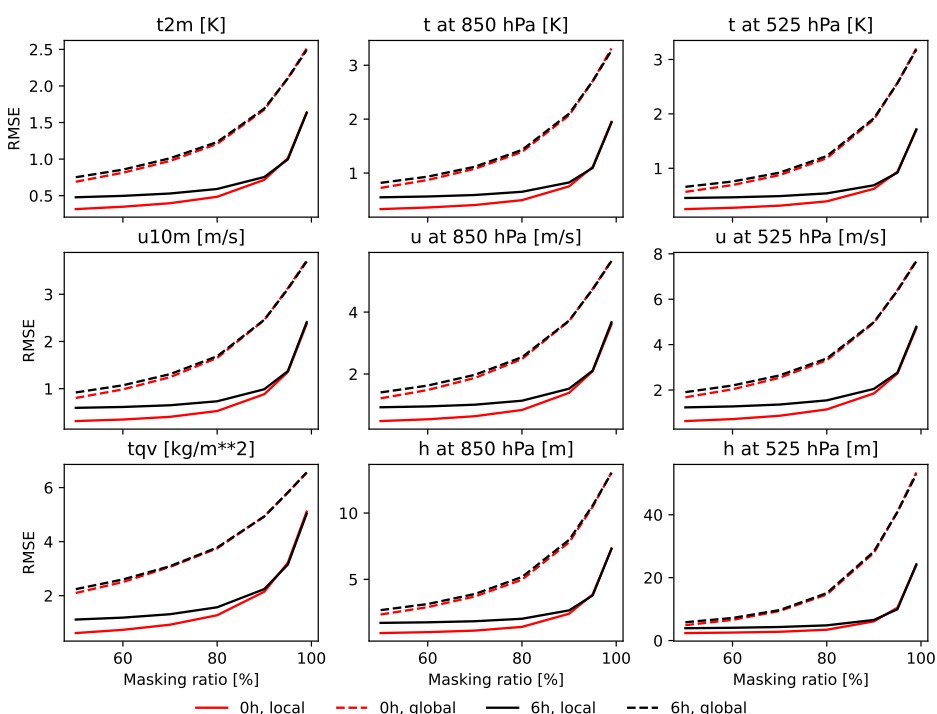

Figure 11: Zero-shot reconstruction performance of Prithvi WxC evaluated with 50, 60, 70, 80, 90, 95 and 99% masking. Note that the 6-hour ahead values are without any forecast tuning.

Figures 11 and 12 give additional results regarding zero-shot reconstruction and forecasting. In particular, both figures show additional variables beyond what was discussed in the main body of the text. The overall narrative however remains the same. Prithvi WxC is highly competitive at short lead times yet forecasting accuracy diminishes more rapidly with lead time than in other models.

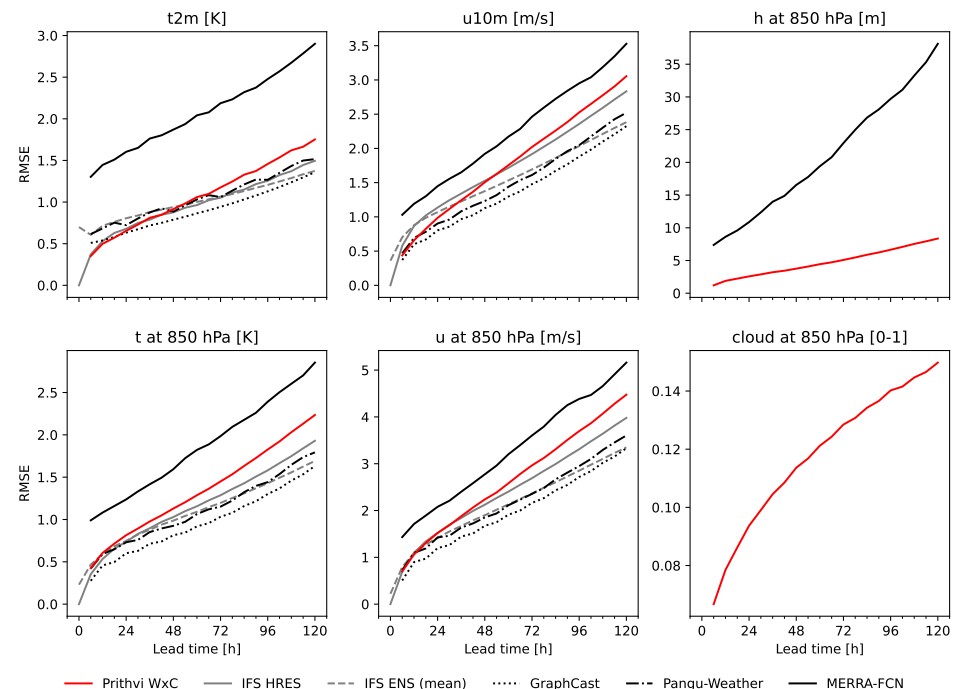

Figure 12: Zero-shot forecasting performance of Prithvi WxC.

## C.2 HURRICANE TRACK FORECASTING

The complete list of hurricanes used for zero-shot hurricane track forecasting is shown in table 10. One significant example is Hurricane Ida, a Category 4 storm that struck Louisiana in 2021. This hurricane, the second-most damaging in Louisiana's history after Hurricane Katrina, is presented as a sample track and intensity in Figure 5. Prithvi WxC demonstrated superior accuracy in both track and intensity predictions. The mean track error for Prithvi WxC was 63.9 km compared to the observed tracks, significantly outperforming the MERRA-2 trained FourCastNet (201.939 km) and the ERA5 trained FourCastNet (262.323 km). Moreover, the Prithvi WxC accurately forecasted both the time and location of Ida's landfall, with a landfall location error of less than 5 km, in contrast to errors greater than 20 km for the other models. Intensity predictions, measured in MSLP and 10-meter sustained wind speed, also favored Prithvi WxC, which outperformed the MERRA-2 trained FourCastNet and showed reasonable consistency with the ERA5 trained FourCastNet. Spatial distribution of Sea Level Pressure (SLP) for a 60-hour forecast (valid for 12 UTC on 2021-08-29) are shown the figure 5 c-e. Among the models, the WxC model predicts the hurricane landfall most accurately in terms of both spatial location and timing, compared to the HURDAT reference.

## D FINE-TUNING VALIDATION

### D.1 DOWNSCALING

Downscaling models are used to refine low-resolution data to provide localized information. Several studies (Doury et al., 2023) (Lessig et al., 2023) (Nguyen et al., 2023a) (Stengel et al., 2020) employ AI models as downscaling emulators to learn the relationship between low-resolution input data and high-resolution output fields. We use a pretrained Prithvi WxC to recover the spatial structure of coarsened near surface temperature for two different datasets - MERRA-2, and CORDEX-CMIP5-RCP8.5 - with different input variables and different input resolutions.

We use the architecture 13 to fine-tune Prithvi WxC for the downscaling task. The patch embedding layer encodes static and dynamic data for surface variables and variables at different pressure levels and optionally for multiple time steps. The first upscaling module is used for shallow feature extrac-

Table 10: List of Hurricanes for Evaluation

| Name (YYYY) | Category | #IC | Initial Conditions |
|---|---|---|---|
| Jose (2017) | C4 | 4 | 2017090900, 2017091000, 2017091100, 2017091200 |
| Harvey (2017) | C4 | 2 | 2017082400, 2017082500 |
| Irma (2017) | C5 | 3 | 2017090500, 2017090600, 2017090700 |
| Michael (2018) | C5 | 2 | 2018100800, 2018100900 |
| Florence (2018) | C4 | 4 | 2018091000, 2018091100, 2018091200, 2018091300 |
| Dorian (2019) | C5 | 4 | 2019083100, 2019090100, 2019090200, 2019090300 |
| Lorenzo (2019) | C5 | 4 | 2019092500, 2019092600, 2019092700, 2019092800 |
| Humberto (2019) | C3 | 2 | 2019091400, 2019091500 |
| Delta (2020) | C4 | 2 | 2020100600, 2020100700 |
| Laura (2020) | C4 | 2 | 2020082300, 2020082400 |
| Iota (2020) | C4 | 2 | 2020111400, 2020111500 |
| Zeta (2020) | C3 | 1 | 2020102500 |
| Eta (2020) | C4 | 2 | 2020110700, 2020110800 |
| Teddy (2020) | C4 | 5 | 2020091400, 2020091500, 2020091600, 2020091700, 2020091800 |
| Ida (2021) | C4 | 3 | 2021082700, 2021082800, 2021082900 |
| Grace (2021) | C3 | 2 | 2021081700, 2021081800 |
| Larry (2021) | C3 | 5 | 2021090200, 2021090300, 2021090400, 2021090500, 2021090600 |
| Sam (2021) | C4 | 5 | 2021092500, 2021092600, 2021092700, 2021092800, 2021092900 |
| Ian (2022) | C5 | 4 | 2022092500, 2022092600, 2022092700, 2022092800 |
| Fiona (2022) | C4 | 4 | 2022091600, 2022091700, 2022091800, 2022091900 |
| Franklin (2023) | C4 | 1 | 2023082200 |
| Lee (2023) | C5 | 8 | 2023090500, 2023090600, 2023090700, 2023090800, 2023090900, 2023091000, 2023091100, 2023091200 |
| Idalia (2023) | C4 | 4 | 2023082700, 2023082800, 2023082900, 2023083000 |

tion for lower frequency components and also used to control the token resolution that is input to the Prithvi WxC model. This follows a deeper feature extraction by the pretrained transformer model. Since we set the masking ratio in the encoder to 0 % and the data becomes dense, we may introduce a Swin-shift in the encoder. Note that we can make this change *while keeping the core transformer layers frozen*. Following (Liang et al., 2021), we use a convolution layer after the transformer to enhance translational equivariance, which is important in downscaling when using different local grids. The residual connection between the shallow and deep feature extraction layer allows combining lower spatial frequency information with the higher spatial frequency information. The final upscale layer focuses on extracting and refining specified output fields.

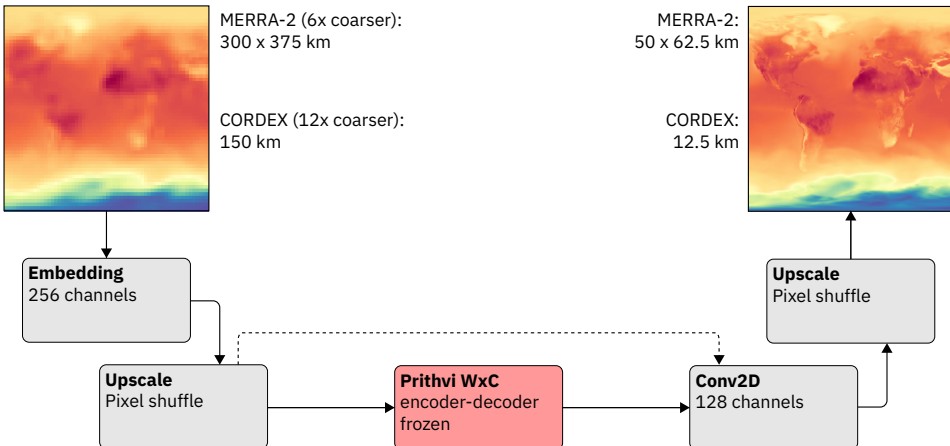

Figure 13: Fine-tuning Architecture of Prithvi WxC for downscaling. The "upscale" blocks before and after the backbone increase the resolution of the data.

**MERRA-2**  As outline in the main text we fine-tune a 6x weather downscaling model for 2m surface temperature using MERRA-2 data. The input data variables are the same as used for pretraining. We first coarsen MERRA-2 data from dimension 361 x 576 (50km x 62.5km resolution) to dimension 60 x 96 (300km x 375km resolution), and secondly apply a smoothing operation in form of a convolution with a 3x3 pixels kernel. Upscaling by a factor 2 before Prithvi WxC we increase the data resolution to 120 x 192 (150km x 187.5km). By using a patch size of 1 for tokenization, we make the token resolution similar to the token resolution that Prithvi WxC model was pretrainedon (100km x 125km). We then upscale by a factor of 3 to restore the low-resolution data to the original resolution of the 360 x 576 (50km x 62.5km). As the power spectra in Figure 7 (c) show, the interpolation baselines are poor at reconstructing the higher frequency wavenumbers of the ground truth, while the fine-tuned Prithvi WxC downscaling model is able to do so.

**CORDEX**  In comparison to the MERRA-2 setting, downscaling CORDEX data changes the dataset, the temporal step as well as the domain from the pretraining case. i.e. we are now tuning the model for a local context on a dataset that comprises significantly less inputs than the original MERRA-2 dataset.

We fine-tune a 12x climate downscaling model for daily mean near-surface air temperature for a period from 2006 to 2100 under scenario RCP8.5 (Moss et al., 2010). All CORDEX input data variables (daily means) are shown in Table 5. Following (Doury et al., 2023), we use a perfect model framework, downscaling coarsened regional climate simulations, rather than training a model to map GCM simulations to RCM simulations which may not be very well correlated. First, we coarsen the input data of dimension 444 x 444 (12.5 km x 12.5 km resolution) to dimension 37 x 37 (150 km x 150 km resolution) and apply a smoothing convolution such as for MERRA-2. One upscaling layer before the Prithvi WxC backbone increases the data resolution by a factor of 3 to a dimension of 111 x 111 (50 km x 50 km resolution) and two upscaling layers after the Prithvi WxC backbone increase resolution by a factor of 2 each to restore the CORDEX data's original 12.5 km x 12.5 km resolution. Model performance is evaluated on data from simulation scenario RCP4.5 which was not seen during training. Similar as for the MERRA-2 downscaling, the power spectra in

Figure 8 (c) demonstrate better reconstruction of higher frequency wavenumbers by the fine-tuned Prithvi WxC downscaling model compared to the interpolation baselines.

## D.2 Climate Model Parameterization for Gravity Wave Flux

This task uses the pretrained Prithvi WxC to create a fine-tuned model for climate applications. The fundamental question being: can we (re-)use large AI models to develop improved, data-driven climate model parameterizations for small-scale atmosphere-ocean processes?

**Background:** Atmospheric gravity waves (GWs) are intermittent, small-scale ($\mathcal{O}(1)$ to $\mathcal{O}(1000)$ km) perturbations generated around thunderstorms, jet disturbances, flow over mountains, etc. (Fritts & Alexander, 2003; Achatz et al., 2023). Gravity waves couple the different layers of the atmosphere by carrying surface momentum to stratospheric and even mesospheric heights. Yet, most climate models fail to resolve them owing to limited resolution. Thus, they belong to a class of key physical processes crucial to the earth's momentum budget but only crudely represented in coarse-climate models using inadequate *physical parameterizations*.

An improved parametric representation of gravity waves in comprehensive climate models can potentially improve the representation of the seasonal transitions (McLandress et al., 2012), clear air turbulence (Plougonven & Zhang, 2014), Antarctic extreme heat (Choi et al., 2024), and tropical predictability (Baldwin et al., 2001); leading to more certain climate predictions and advancements in mechanistic understanding.

From an AI perspective, this downstream prediction task moves from predicting the large-scale atmospheric state prediction to smaller-scale state prediction, and leverages the cross-scale learning from pre-training. As such, the finetuning task is defined to use the latent space of Prithvi to develop data-driven physical parameterizations to provide missing sub-grid scale variability in coarse-climate models at zero-lag. This is somewhat akin to the downscaling task where CORDEX is used to augment missing sub-grid information during fine-tuning. For this task, the model is fine-tuned using high-fidelity, high-resolution gravity wave data extracted from ERA5 (which resolves a substantial portion of the atmospheric gravity waves, if not all).

### D.2.1 Extracting GW data for finetuning.

The goal is to accurately predict the momentum fluxes carried by waves generated in different parts of the globe by different processes, given the background atmospheric state. The approach is similar to that followed by traditional single-column parameterizations (Lott & Miller, 1997; Scinocca, 2003; Kim et al., 2003). Here, we do so by learning from high-resolution data. In very simple terms, given the background atmospheric state around a mountain (e.g., Andes), or around tropical storm, can our ML model predict whether the waves are spontaneously generated, and if they are, calculate the net momentum fluxes they carry; not unlike predicting the cloud cover for a given set of atmospheric conditions.

We use four years of ERA5 global reanalysis on 137 model vertical levels and 30 km horizontal resolution at hourly-frequency to prepare the training data for fine-tuning. The top 15 levels, i.e., levels above 45 km are removed due to artifical sponge damping in effect, so effectively 122 vertical levels. The model takes the zonal wind speed ($u$), meridional wind speed ($v$), the temperature ($T$), and pressure ($p$), along with positional variables latitude, longitude, and surface height as input. These variables collectively describe the background state of the atmosphere. The model outputs the directional momentum fluxes carried by gravity waves. These fluxes describe the net instantaneous momentum the gravity waves carry. These directional fluxes are mathematically expressed as the covariances ($u'\omega', v'\omega'$), and are computed using Helmholtz decomposition using the horizontal ($u,v$)=(U,V) and vertical wind speeds ($\omega$=OMEGA). Both the input and output are conservatively coarse-grained to a $64 \times 128$ ($\approx 300$ km) latitude-longitude grid to be consistent with a typical coarse-climate model and to remove phase dependencies of the calculated fluxes.

### D.2.2 Finetuning Prithvi WxC

The architecture schematic for the finetuning is shown in Figure 14. During fine-tuning Prithvi WxC, we freeze the encoder and decoder part of the model. The frozen encoder is preceded by 4 learnable convolution blocks each with an increasing number of hidden channels, i.e., $C$, $2C$, $4C$ and then

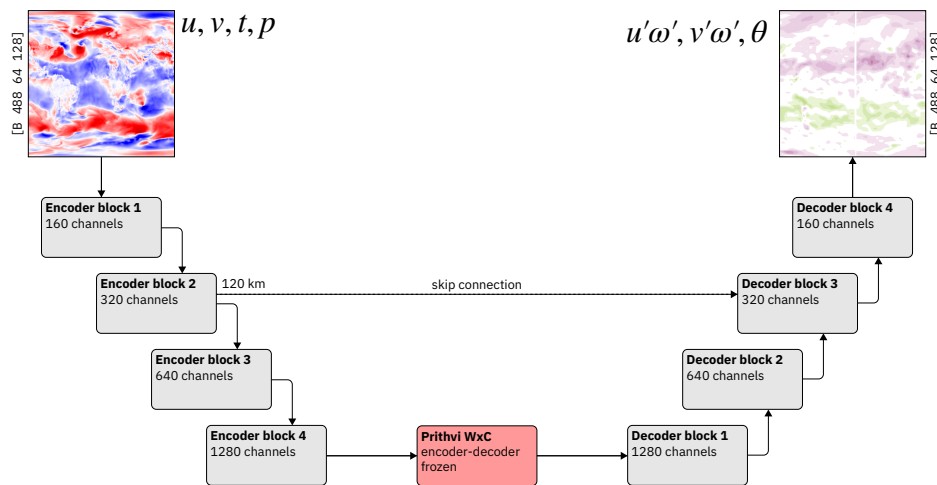

Figure 14: Finetuning Architecture of Prithvi WxC for parameterization of gravity wave flux

$8C$, where $C = 160$. Likewise, the frozen decoder is succeeded by 4 new learnable convolution blocks. Since gravity wave flux prediction is an instantaneous flux calculation task, we fix the lead time $\delta t$ to zero. The instantanous model input for fine-tuning has shape [1, 488, 64, 128] where the 488 channels comprise the four background variables $u$, $v$, $t$ and $p$ on 122 vertical levels each, and on a $64 \times 128$ horizontal grid, as discussed above. The model was fine-tuned to produce an output with shape [366, 64, 128] comprising of the potential temperature, $u'\omega'$, and $v'\omega'$ on 122 vertical levels each.

The fine-tuning model leveraged a U-Net like architecture to allow the model to extract high-frequency information from the given data source. We re-emphasize that Prithvi WxC was pre-trained on the MERRA-2 dataset but for fine-tuning we are using the downscaled ERA5 dataset. More importantly, the finetuned model uses global information as input to predict global fluxes as output, providing a direct contrast to traditional single-column parameterizations. Access to global information allows the model to learn the horizontal propagation of gravity waves.

