# OpenReview forum: "A Foundation Model for Weather and Climate"
_ICLR.cc/2025/Conference — Submitted to ICLR 2025_

### Official Review · Reviewer_j9By · 2024-10-15

**Soundness:** 3
**Presentation:** 3
**Contribution:** 3
**Rating:** 6
**Confidence:** 3

**Summary:**

The paper titled "Basic Models for Weather and Climate" introduces Prithvi WxC, a 2.3 billion parameter basic model for various weather and climate tasks. These include downscaling, autoregressive prediction, and extreme event estimation. The model is trained on 160 variables from the MERRA-2 dataset and combines mask reconstruction with prediction tasks to learn from various atmospheric data. Its encoder decoder architecture and ability to work with different spatial topologies make it suitable for global and regional weather modeling.

**Strengths:**

Originality: The introduction of basic models for weather and climate applications is novel and important. Unlike specific task models such as FourCastNet and GraphCast (Lam et al., 2022), Prithvi WxC addresses a wide range of tasks and effectively narrows the gap between artificial intelligence models for specific weather tasks and general artificial intelligence base models.
Quality: The mixed pre training objective of this model combines masking and prediction, which is robust, especially because it uses climate bias rather than just future state prediction, which enhances the adaptability of the model. The model also showed impressive results in zero sample evaluation of reconstruction and autoregressive prediction, outperforming the baseline in a short delivery cycle.
Meaning: The ability to generalize to multiple downstream tasks, such as downscaling and gravity wave parameterization, suggests that this model has the potential to have a significant impact on weather and climate modeling.

**Weaknesses:**

Generalization to other datasets: Although the model performs well on MERRA-2 data, its generalization ability to other datasets such as ERA5 or CMIP has not been fully explored. Validation on different datasets will better demonstrate its robustness.
Long term prediction: The accuracy of Prithvi WxC decreases with the extension of the prediction window, especially beyond 66 hours, and its performance is poor compared to models such as Pangu. A deeper investigation into how to maintain performance within an extended time frame can improve its utility in medium - and long-term forecasting.

**Questions:**

Architectural flexibility: Can you provide a detailed explanation of the adaptability of the architecture when applied to non rectangular grid systems, such as those used for ocean simulation or polar regions?
Generalization strategy: What steps are taken to ensure the performance of the model when training or fine-tuning on datasets outside of MERRA-2, such as ERA5 or even higher resolution datasets?
Hurricane forecast: Can you provide more details on how the hurricane trajectory prediction of this model compares to specific task models such as FourCastNet, especially in areas with sparse data coverage?

---

### Official Review · Reviewer_noFm · 2024-10-26

**Soundness:** 3
**Presentation:** 3
**Contribution:** 2
**Rating:** 3
**Confidence:** 4

**Summary:**

The paper introduces a new foundation model for weather and climate, called Prithvi WxC, with 2.3 billion parameters, and trained on relatively unconventional yet interesting reanalysis products of MERRA-2. The authors use a relatively novel pre-training strategy in this field, where in addition to forecasting-only pre-training, they also combine masking for reconstruction in the hope of better self-supervision, among several upsides (e.g., natural extension to data assimilation with sparsely-gridded observation). The FM is then evaluated on several downstream tasks, including forecasting, downscaling, and parameterization.

**Strengths:**

Operationally, the large-scale pre-training of 2.3B FM is impressive in the emerging field of AI4Science. The use of large-scale dataset is also noteworthy. The writing is clear and the downstream tasks are clearly defined, and touch upon some of the hardest challenges facing the field. The use of beyond-atmospheric variables (coupled ocean + land) is also welcomed to build a full Earth system FM.

**Weaknesses:**

There are several major weaknesses in the paper, including non-existent/weak baselines and misleading claims, summarized below:

Major weaknesses
1. The term foundation model for this work is misleading, since in contrast to other similar works e.g., ClimaX, Aurora, the model is only trained on one data source which is MERRA-2. The problem: reanalysis products are by construction applicable for short-medium range applications as they are used either to evaluate NWPs or used as ICs for the next forecasting window. The climate dataset being missing, makes the claim of Prithvi being also a climate FM is an overreach at best.

2. Related to the first point, the paper does not evaluate on any climate-related tests despite claiming it to be a climate FM too: (a) is the model stable over 50-100 years climate rollout? (b) what is the climate drift/bias compared to SOTA climate emulator? The paper applies downscaling to a climate dataset CORDEX, but this is less of a climate question and just a general downscaling problem since the former is more concerned about getting the long-term statistics, rather than a high-resolution state realization, correct (which is near impossible to nonlinear chaotic systems such as the Earth system).

3. The downscaling benchmarking is also unacceptable as the baseline is too weak (e.g., bilinear, nearest neighbor). The author should either remove this part or add stronger baselines where the SOTA is at least a deep learning based method. Also, there is no benchmarking on the parameterization downstream task. At least use existing DL-based models from recent works.

4. The forecasting performance appears to be unconvincing at best: with 2.3B parameter, it performs worse than e.g., <50M parameter GraphCast (~40x smaller) even at short lead time of 66 hours (<3 days). Even when the authors mention this result is "zero-shot" (which I find it unconvincing since there is rollout fine-tuning still) and the target is different (MERRA-2 vs ERA5), the obvious larger error growth (Figure 4) is alarming as it may not be useful for long-range climate forecasting. Also, why not benchmark against MERRA-FourCastNet in the forecasting task since this provides for a fairer comparison as both are trained with MERRA (as in the case for hurricane prediction). Finally, figure 4c (single line: forecasting cloud) is a case-in-point: the lack of sufficient apple-to-apple baselines where training/eval is done on MERRA-2.

Overall, I find the results unconvincing given the lack of data sources, proper baselines, and inferior performance gain despite Prithvi being orders-of-magnitude larger than any SOTA model. As a side note: I believe the task of downscaling and parameterization is similar in that both attempts to resolve small-scale physics in an otherwise coarse-resolution model. I suggest the authors combine or use different downstream tasks e.g., climate projection.

**Questions:**

In addition to the weaknesses above, can the authors clarify the following:

1. How is the masking pre-training strategies better than just using variable lead-time that includes delta_t = 0?
2. How is the local-global attention better than existing pre-training strategies of e.g., patch-variable-level tokenization employed in e.g., ClimaX?
3. With masking as an additional pre-training strategy, what is the cost-performance tradeoff?
4. Has the authors measure Prithvi's parameterization stability in an online-coupled setting?

---

> ### Author Response · Authors · 2024-12-02
> **Response to questions raised**
>
> We would like to thank the referee for their thoughtful feedback. This post contains responses to the questions raised, the next will address perceived weaknesses. Please consider the updated draft with changes as follows:
> - New appendix (B.2) including ablation studies on the use of climatology, the impact of the masking strategy, handling of the lead time signal, the use of context tokens and scaling performance.
> - New baseline for forecasting results: A version of FourCastNet trained on MERRA-2 data. (Figures 4 and 11.)
> - Typos, clarifications and references added in response to referee pzCZ.
>
> **Q1** From a conceptual point of view, the case $\delta t = 0$ becomes trivial with a non-masking objective. This does not hold only for our training objective, but also for a series of other, standard objectives. If the objective is that of "tendencies" as done by e.g. GraphCast (predicting $X_{t+ \delta t} - X_t$ from $X_t$), the optimal model predicts $0$ for $\delta t = 0$. If on the other hand you generate absolute quantities ($X_{t+ \delta t}$ from $X_t$), the optimal model for $\delta t = 0$ is an identity operator. In our case where the model effectively generates anomalies, the model only learns to compute $X_{t+\delta t} - C_{t+\delta t}$ when $\delta t = 0$. Naturally this is just motivation, so please refer to our new ablation studies in section B.2.1, especially the paragraph ``the effect of masking'' and table 7. As you will see, the reconstruction performance falls off the cliff when only training without masking. And we expect reconstruction performance to be crucial when looking at use cases such as the work by Vandal et al. [1] and McNally et al. [2]. Note that neither [1] nor [2] comment too much on their masking approaches. See however appendix D.1 in [1] where the authors report a masking ratio as extreme as 98.6%.
>
> **Q2** The local-global attention allows us to scale to high token counts. ClimaX [3] ran on a resolution of 1.40625 degrees and coarser. This is the same resolution as ORBIT [4] and STORMER [5]. While we cannot speak for the authors of those papers, this choice of resolution is presumably due to vanilla ViT architectures not scaling to sufficiently high token counts to work with data of MERRA-2 or ERA5 resolution at reasonable patch size. To our knowledge every model does one of the following: work at coarse resolution (ClimaX, ORBIT, Stormer), train on regional patches (AtmoRep), use non-vanilla ViT architectures such as Swin Transformer (Pangu-Weather [5], Aurora [6]) or use custom architectures such as GraphCast or FourCastNet. (When writing vanilla ViT we strictly refer to the question of scaling of the attention computation; there are many great innovations such as variable aggregation). So for us, the true comparison point is the Swin transformer of Pangu-Weather and Aurora rather than ClimaX. Swin transformers are highly performant, yet there are two drawbacks. One is that there is no way to do memory efficient masking. (You could mask tokens but that negates the memory effects.) The other is that using a Swin transformer always restricts to a rectangular structure of the data. We could have chosen to use a Long-short transformer, yet that comes with similar tradeoffs. The local-global attention allows us to push to high resolutions while not imposing strong constraints on topology while masking.
>
> **Q3** Thanks for raising this point. Please see appendix B.2.1, Table 7 for detailed results as well as remarks made for question 1.
>
> References
> - [1] Vandal, Thomas J., et al. "Global atmospheric data assimilation with multi-modal masked autoencoders." arXiv preprint arXiv:2407.11696 (2024).
> - [2] McNally, Anthony, et al. "Data driven weather forecasts trained and initialised directly from observations." arXiv preprint arXiv:2407.15586 (2024).
> Zhu, Chen, et al. "Long-short transformer: Efficient transformers for language and vision." Advances in neural information processing systems 34 (2021): 17723-17736.
> - [3] Nguyen, Tung, et al. "ClimaX: A foundation model for weather and climate." arXiv preprint arXiv:2301.10343 (2023).
> - [4] Wang, Xiao, et al. "Orbit: Oak ridge base foundation model for earth system predictability." arXiv preprint arXiv:2404.14712 (2024).
> - [5] Nguyen, Tung, et al. "Scaling transformer neural networks for skillful and reliable medium-range weather forecasting." arXiv preprint arXiv:2312.03876 (2023).
> - [6] Bi, Kaifeng, et al. "Accurate medium-range global weather forecasting with 3D neural networks." Nature 619.7970 (2023): 533-538.
> - [7] Bodnar, Cristian, et al. "Aurora: A foundation model of the atmosphere." arXiv preprint arXiv:2405.13063 (2024).
> - [8] Lam, Remi, et al. "GraphCast: Learning skillful medium-range global weather forecasting." arXiv preprint arXiv:2212.12794 (2022).
> - [9] Pathak, Jaideep, et al. "Fourcastnet: A global data-driven high-resolution weather model using adaptive fourier neural operators." arXiv preprint arXiv:2202.11214 (2022).

---

> > ### Author Response · Authors · 2024-12-02
> > **Response to perceived weaknesses**
> >
> > Once again we would like to thank the reviewer for their thoughtful remarks. As suggested before, this post continues our response by addressing perceived weaknesses.
> >
> > **W1 & W2** It is probably best to address these together as the questions focus on the characterizatio of Prithvi WxC as an FM for weather _and_ climate vs. one for weather alone. Let us start by responding to W2, the question about long term rollouts. Our original objectives do not include long term rollouts and there is no place in the paper where we make a claim that the model is suitable for that. Naturally one might try, but given that we did not optimize the model in any way for stability here it is unlikely to be stable under such time ranges. From our point of view, half of the downstream tasks in the paper are relevant in a climate context, namely the parameterization problem as well as the downscaling of CORDEX data. Which is why we chose this label. Now, concerning the issue of datasets in W1, there was a similar, yet not identical, point raised by reviewer HFKb. We would ask you to consider that response as well for details. Yet the gist of our argument is that while we agree that training on multiple datasets is most likely useful, as shown by e.g. the Aurora results, it is not necessary. What is necessary is to show performance on a number of different datasets, different resolutions, different time steps etc. Whether that performance is obtained zero shot, via tuning to that dataset or via pretraining on multiple datasets is a secondary concern. Our paper uses three different datasets (ERA5, MERRA2 and CORDEX) at different resolutions and time steps. Admittedly the point you are raising is a bit more subtle, namely that reanalysis datasets have specific uses. Yet consider again the previous part of this answer that we are not aiming for 100 year rollouts.
> >
> > **W4** We have added the numbers for MERRA2-trained FourCastNet to the graph. Please refer to the updated Figure 4. You will find that we not only outperform the model significantly, MERRA2-trained FourCastNet also shows the same slope. We agree with you that the slope is a point of concern and it is something that we are investigating, but please note that the MERRA2-trained FourCastNet raises the possibility that this is something dataset specific. Regarding the fact that the slope suggests that the model is currently not relevant for long range climate forecasting, let us repeat that we never made the claim that it is to be used for that. And you are correct that the slope seen here is one of the reasons why this is a concern. Yet let us also point out that the short term behavior as well as the reconstruction performance open up tantalizing new use cases such as fine-tuning for data assimilation.

---

### Official Review · Reviewer_HFKb · 2024-11-03

**Soundness:** 2
**Presentation:** 3
**Contribution:** 3
**Rating:** 5
**Confidence:** 4

**Summary:**

The paper presents Prithvi WxC, a new foundation model designed to support a wide range of weather and climate applications. Built on a large, transformer-based architecture, Prithvi WxC is trained on the extensive MERRA-2 dataset, which covers 160 variables capturing atmospheric data. The model is unique in its ability to address multiple tasks—including forecasting, downscaling, and parameterization—making it versatile in handling both regional and global weather patterns.

**Strengths:**

### Originality and significance
Prithvi WxC pushes the foundation model concept in atmospheric science further by expanding beyond just forecasting— seen in earlier models foundation model Aurora. Its architecture and training approach enable it to tackle a variety of downstream tasks, such as downscaling and parameterization, making it a valuable tool for both short-term weather predictions and long-term climate modeling. With its modular design, Prithvi WxC can be adapted flexibly to new tasks that combines AI with physical climate models.


### Quality
The authors have thoroughly evaluated Prithvi WxC across a range of tasks, including zero-shot reconstruction, downscaling, and extreme event forecasting. The extensive validations across different downstream tasks support Prithvi WxC’s adaptability and effectiveness in diverse weather and climate applications.



### Clarity and open research
The paper is organized in a clear, logical flow, moving smoothly from motivation and background to model architecture, objectives, and results. Key ideas, like the mixed masking and forecasting objective, are presented in a way that makes the technical contributions accessible to both AI and climate science audiences. Code and comprehensive supplementary materials are provided.

**Weaknesses:**

### Reliance on a single reanalysis dataset (MERRA-2)
MERRA-2, with its relatively lower spatial resolution, is not commonly used in AI-driven weather and climate research, where higher-resolution datasets like ERA-5 are preferred for their superior predictive accuracy. The authors themselves acknowledge this limitation, citing the weaker performance of Prithvi WxC in hurricane track forecasting compared to the ERA-5-trained FourCastNet, attributing this discrepancy to MERRA-2's lower spatial resolution.

Prithvi WxC’s exclusive training on the MERRA-2 reanalysis dataset raises questions about whether it truly qualifies as a foundation model. The narrower training base implies that the model may be learning a representation more specific to MERRA-2 characteristics, along with its biases and errors, rather than capturing a broader, more generalized understanding of weather and climate dynamics. By contrast, Aurora foundation model are pretrained on six diverse weather and climate datasets, including ERA-5, CMCC, IFS-HR, HRES Forecast, GFS Analysis, and GFS Forecasts, which span various sources like forecasts, analyses, reanalyses, and climate simulations [1]. This multi-source approach ensures a broader, more representative foundation that enhances versatility across diverse applications.Prithvi WxC would benefit from a similar multi-dataset training approach to strengthen its robustness and generalizability, which would then qualify it as a foundation model.

[1] Bodnar, C., Bruinsma, W. P., Lucic, A., Stanley, M., Brandstetter, J., Garvan, P., ... & Perdikaris, P. (2024). Aurora: A foundation model of the atmosphere. arXiv preprint arXiv:2405.13063.

### Ablation Study
While the authors propose a novel objective function, they only hypothetically attribute Prithvi WxC's strong short-term forecasting performance to its masking objective, without providing empirical evidence. This lack of testing weakens the claims about the model’s unique architecture and objective function. The observed performance could be influenced by several factors: the mixed objective itself, specific network structures or attention mechanisms, and choices made in the pretraining setup.

Without ablation experiments, the paper's assertions about the effectiveness of these innovations remain speculative, leaving readers uncertain about the impact of each component. An ablation study could isolate the contributions of these elements and would strengthen the paper by making its claims more concrete and providing clearer insights into Prithvi WxC’s architectural and training contributions.

**Questions:**

### Minor issue:  Weak baselines for downscaling
The comparison primarily involves interpolation-based methods and does not consider more advanced, AI-driven downscaling models or domain specific statistical/dynamical downscaling.

---

> ### Author Response · Authors · 2024-11-26
> **New version & responses to reviewer**
>
> We would like to thank the reviewer for their clear and thoughtful feedback and concise remarks. We just uploaded a new version of the paper which includes
> - A new baseline for the forecasting section.
> - Detailed ablation studies in appendix B.2
> - Various minor edits to address typos and make clarifications.
>
> Regarding the issues raised by the reviewer:
>
> **Reliance on a single reanalysis dataset (MERRA-2)**
>
> We very much agree that a _foundation model_ in this domain should be useful across a variety of datasets. In addition, we would argue that models should be applicable to a range of resolutions, time scales and crucially also multiple regions (which partially motivates our masking approach). Also, the strong results of Bodnar et al. in Aurora indeed suggest that using multiple datasets during pretraining is a great way to achieve applicability to multiple datasets.
>
> Having said that, the defining criteria of the foundation model is probably _applicability_ to a range of datasets, not what it was pretrained on. And indeed, while we chose not to pretrain on multiple datasets, we purposely evaluated on a variety thereof to show relevance beyond MERRA-2. Apart from MERRA-2, our work uses ERA5 (gravity wave parametrization) and CORDEX (downscaling). So our work does include a variety of datasets and tests the model's performance beyond MERRA-2. Along this line, please note the work of Subich (reference below) which includes a careful discussion of how to fine-tune GraphCast to a different dataset. So in the same way that vision or language models can be tuned to new datasets, models of physical systems can (within reason) be tuned to new datasets.
>
> Finally, we do not only tune to different datasets, but also different resolutions, temporal steps (both on the input and output side) and different regional settings. We are not aware of any prior work varying all these parameters.
>
> Reference:
> Subich, Christopher. "Efficient fine-tuning of 37-level GraphCast with the Canadian global deterministic analysis." arXiv preprint arXiv:2408.14587 (2024).
>
> **Ablation Study**
>
> Thank you for making this point. We added a new appendix (B.2) giving detailed results of our ablation studies and the resulting choices regarding architecture and pretraining objective. Regarding the latter, please note the results regarding both the choice of climatology as well as the impact of masking.

---

> > ### Comment · Reviewer_HFKb · 2024-11-30
> > **Response to authors**
> >
> > Thank you for your thoughtful and detailed response!
> >
> > I completely agree that a foundation model should be versatile across datasets, resolutions, time scales, and regions. Your approach of tuning across these dimensions is impressive and clearly addresses some important aspects of generalization.
> >
> > That said, I still believe that pretraining on diverse datasets is critical for building a truly robust foundation. While evaluating on multiple datasets like ERA5 and CORDEX is definitely valuable, pretraining on diverse datasets helps create a deeper, more generalized representation that can better handle the biases and limitations of any single dataset. We've seen this in NLP and computer vision, where multi-dataset pretraining has been key to achieving state-of-the-art performance and adaptability across tasks.
> >
> > I also appreciate the mention of fine-tuning, as it’s a great way to adapt to new datasets. But I think broader pretraining reduces the need for heavy fine-tuning, especially in situations where labeled data or computational resources are limited.
> >
> > Finally, the flexibility of your model to handle different resolutions, temporal steps, and regions is really impressive. I see this as a great complement to multi-dataset pretraining rather than a replacement. Combining both strategies would make for an even stronger foundation model that’s truly adaptable to a wide range of scenarios.
> >
> > I hope this adds to the conversation and helps push the field forward. Thanks again for engaging so thoroughly!

---

> > > ### Author Response · Authors · 2024-12-01
> > >
> > > Thank you for your valuable feedback, it was really nice to engage in discussion as well. We will surely consider your comments related to the dataset for future work.  If you are satisfied with the work and advancements, we hope that you will consider increasing your rating.

---

### Official Review · Reviewer_pzCZ · 2024-11-04

**Soundness:** 2
**Presentation:** 3
**Contribution:** 3
**Rating:** 5
**Confidence:** 4

**Summary:**

This paper introduces a new foundation model, Prithvi WxC, for atmospheric modeling applications in weather and climate.  Prithvi WxC was trained on 3-hourly data from 1980 to 2019 from the MERRA-2 reanalysis dataset based on a masked reconstruction/forecasting pre-training objective.
The model follows a transformer-based encoder-decoder architecture inspired by Hiera and MaxViT.
Afterward, the model is fine-tuned for various downstream tasks: Medium-range weather forecasting, global and regional downscaling, and learning a gravity wave flux parametrization. These tasks have different sets of spatial resolutions, variables, and datasets, showing the flexibility of the foundation model.

**Strengths:**

1. Prithvi WxC is quite flexible as it can be used for a broad range of downstream applications, as convincingly shown in the experiments.
2. The method contains original ideas such as the pre-training objective and using climatology-derived anomalies as targets, which I found interesting to read about.
3. The paper is generally clearly written and easy to read.

**Weaknesses:**

1. The paper falls short of establishing a compelling case for Prithvi WxC as a foundation model for weather or climate. The practical significance and advantages of this approach remain inadequately demonstrated:

a.) While foundation models typically excel at zero-shot performance and data-efficient fine-tuning across diverse tasks, the evidence presented for Prithvi WxC's capabilities in these areas is not convincing. Baselines for the non-forecasting experiments are either very weak (interpolation-based downscaling) or non-existent (gravity wave experiments). Some highly relevant and simple baselines are:
- How much worse(?) does Prithvi WxC perform on these tasks if you omit the pre-training stage (i.e. initialize with random weights instead of the frozen pre-trained ones, and train all parameters jointly from scratch on the tasks)?
- How about completely removing the pre-trained transformer backbone (i.e. removing the Prithvi WxC block from Figures 12 & 13)?
- For the latter, it would be also good to run an experiment where you replace the pre-trained Prithvi WxC backbone with some "lightweight" blocks (e.g. a (deeper) U-Net), trained in a task-specific way from scratch, to account for the huge difference in parameter counts if you completely remove Prithvi WxC.

These ablations would immensely help in understanding how useful the pre-training stage is for these downstream applications (e.g. does using pre-trained Prithvi WxC improve performance over such simple baselines? Is it more data-efficient?). Besides, otherwise, it is hard to see evidence for the claim in the conclusion that *"Instead of building task-specific ML-models from scratch, these pretrained encoders can be used to develop more precise data-driven models of atmospheric processes"*.

b.) No ablations are included. I understand that training such a huge model is expensive but having a few ablations would have been very appreciated (perhaps, with a smaller-scale version of the model). For example:

- How crucial is it to predict climatology-normalized targets as opposed to normal per-variable means/stds?
- What's the forecasting performance of Prithvi WxC after the first pre-training phase?
- How important is local vs global masking? What about the masking rates?
- What's the line of thought behind randomizing the distance between input timesteps? Can the model only use one input timestep? I presume this is possible by masking the corresponding snapshot by 100%, but no experiments with this setting are shown.

c.) The weather forecasting results seem lukewarm, albeit it is hard to judge because the comparison is not apples-to-apples.
- Prithvi WxC is trained and evaluated on Merra-2. The baselines are evaluated on ERA5. These reanalysis datasets have different spatial resolutions. The evaluation years seem to be different too (correct me if I'm wrong). It would help to fix this mismatch. For example,  given the foundational nature of Prithvi WxC... why not fine-tune it on ERA5 directly? Showing that it can be competitive to these baselines in an apples-to-apples comparison would be a very strong result.
- Based on the mismatched comparison, Prithvi WxC seems to be competitive on 6h to 12h forecasts but it's quite notable that its performance implodes compared to the baselines for longer lead times. It is very unclear why. I wouldn't necessarily expect this version of Prithvi WxC to be state-of-the-art, but the performance does seem underwhelming. Especially given that the authors did "several things" to tune these results (i.e. a second forecasting-specific pre-training stage and autoregressive rollout fine-tuning).
- The hurricane evaluation includes hurricanes from 2017 to 2023. This seems to overlap with the training data period (up to 2019).
- Either Figure 6 or its analysis in the main body of the text (lines 251-253) is wrong because I see all of the three models do best on exactly one of the three RMSE figures.
- For the hurricane forecasting experiments, I would appreciate a comparison to the state-of-the-art models included in the weather forecasting experiments (e.g. GraphCast) which have shown to be better than FourcastNet.

d.) The downscaling problem setup is artificial. Downscaling coarsened of existing reanalysis/model outputs is not of much use in practice. A realistic and important downscaling application, as discussed in the Appendix, would be to downscale coarse-resolution model outputs to high-resolution outputs (either of a different model, observations, or the same model run at higher resolution).

e.) The climate model parameterization experiments should be more carefully interpreted.
- The model predicts outputs that are normalized by the 1980-2019 climatology. Unfortunately, decadal or centennial simulations of the future under a changing climate are inherently a non-stationary problem. It is highly unclear if Prithvi WxC would remain stable, let alone effective, under this highly relevant use case. This is particularly so as the in-the-loop (coupled to a running climate model) stability of ML-based climate model parameterizations is a well-known issue.
- The selling point for ML-based emulators of climate model parametrizations is often their computational cheapness. Thus, the runtime of Prithvi WxC should be discussed. Given the large parameter count of Prithvi WxC it might be important to note its runtime as a limitation for these kinds of applications.
- Line 461 claims that Prithvi WxC "outperforms" task-specific baselines but no baselines whatsoever are included in the manuscript for this experiment.
- Are the inputs a global map? I am not familiar with gravity waves, but I believe that most physics parameterizations in climate models are modeled column-wise (i.e. across atmospheric height but ignoring lat/lon interactions). This is surely a simplification of these parameterizations, but it seems to indicate that they're highly local problems. What's the motivation for using global context then?
- The end of the section should be worded more carefully, clearly stating the aforementioned limitations.

f.) No scaling experiments are included. Thus, it is unclear how important its 2.3 billion parameter size is, how well the model scales, and how its size impacts performance on the downstream applications. Besides, vision and language models are usually released with multiple model sizes that cover different use cases (e.g. balancing inference speed with accuracy). It would be really useful to get these (and carefully compare them) for Prithvi WxC.

2. Related work is insufficiently discussed. Please include an explicit section discussing it, focusing on:
- Carefully comparing similarities/differences to existing weather foundation models (e.g. architectures, pre-training objectives, downstream applications etc.). Besides, ClimaX is not properly discussed in the paper. Given that it's also a transformer-based foundation model, validated on forecasting, downscaling, and climate emulation, it is very important to include it in the comparison.
- Similarly, please discuss how exactly the masking technique in this paper relates to the ones proposed in Vandal et al. and McNally et al..
- Carefully discuss how the architecture is derived from Hiera and/or MaxViT (and other papers of which components were derived, if any).

3. While the authors transparently discuss some issues/limitations with their experiments (e.g. the evaluation data mismatches), it would be nice to also include an explicit paragraph or section on this (and include aforementioned things like the issues with the climate model parameterization experiments).

Minor:
- Can you properly discuss, and include a reference to, what a Swin-shift is?
- Similarly, for the "pixel shuffle layers"
- Line 39: Pangu -> Pangu-Weather
- Line 48: Nowcasting should be lower-case
- Equation 1: Consider reformulating this as an objective/loss function.
- Also Eq. 1: What is $\hat{X}_t$? What is $\sigma_C$?
- Line 93: $\sigma^2_C = \sigma^2_C(X_t - C_t)$ doesn't make sense to me.
- Line 104: *" same 20 year period that we used for pretraining."* .... Do you mean 40 year period? If not, which 20-year period from the 40-year training period did you use?
- Line 157: Multiple symbols are undefined (e.g. $V_S$).
- Line 169: It's not entirely clear what "alternates" means in this context.
- Line 429: "baseline"... do you mean Prithvi WxC?
- Line 507: "improved"... improved compared to what?
- Figure 12: Do you mean 'downscale' on the right "upscale" block?
- Sections D. 2.3 and D.2.4 in the appendix are literal copies of the corresponding paragraphs on pages 8 and 9. Please remove.

**Questions:**

Major:
- Do you compute area-weighted RMSEs? If not, I strongly encourage fixing this (especially for the weather forecasting experiment; see e.g. Weatherbench2).
- For the fine-tuning experiments, do you start them based on the weights resulting from the first or second pre-training stage? If the former, then the second pre-training seems to also be a form of fine-tuning and I would suggest avoiding using the term "zero-shot" for reporting the forecasting results.
- From looking at Figure 3, I don't think that I agree with the author's interpretation that *"It is interesting that reconstruction performance is relatively little affected by lead time at the lower end of masking ratios"*. There's a clear cap between "0h, global" and "6h, global", especially for the lower end of masking ratios. Do I miss something?
- Can you include downscaling results on more variables than only T2M?

Minor:
- What's the difference between the encoder and decoder blocks? Fig. 1 suggests these are identical... Is the difference some densification by the "reconstruct batch" module in Fig. 1? If so, can you explain this more and make it clearer?
- Line 174: "reduce the masking ratio to 50%"... is this a typo (you use the same rate for the first stage)? What's correct?
- Why is there no reference line in figure 5b)?
- Please define what's meant by spatial and temporal RMSEs.
- Can you include snapshots like Fig. 9 but at the native temporal resolution used for prediction (i.e. not a monthly mean)?
- How does using different patch sizes (larger than the used size of 1) impact downscaling performance?
- Why do you call the model Prithvi WxC?

---

> ### Author Response · Authors · 2024-11-26
> **New version & responses to reviewer**
>
> We would like to thank the reviewer for their highly detailed feedback and thoughtful remarks. Both are much appreciated. We just submitted a new version of the paper which includes an additional baseline and -- crucially -- detailed ablation studies regarding our architecture choices. Regarding the issues raised
>
> 1a) **Baselines** We added a new baseline for forecasting performance. Namely the MERRA2-trained version of FourCastNet.
>
> 1b) **Ablations** We added a new appendix with detailed results from ablation studies showing the impact and tradeoffs of our choices regarding architecture and pretraining objective. This addresses
>   - Effect of climatology.
>   - Effect of masking.
>   - Impact of rollout phase.
>   - Also, note that we randomized the distance between input timestamps as different datasets used have different time steps. Case in point is the CORDEX data which has a daily time step while MERRA2 as used in the paper is 3-hourly etc.
>
> 1c) **Forecasting**
>   - The evaluation years are identical across all models (2020). We added a missing statement to this point to the text.
>   - Tuning the model to ERA5 is a good suggestion and possible future work. However, note that not-quite-apples-to-apples comparisons are not uncommon. I.e. WeatherBench2 itself lists both comparisons against ERA5 as well as the ECMWF HRES analysis. The reason of course being that all models should be compared to the correct ground truth data. Given the lack of MERRA2-based models, it seems good to add some of the best forecast emulators (i.e. GraphCast) here. Note also that we included the MERRA2-trained FourCastNet model as additional baseline and reference point.
>
> 1d) **Downscaling** You are correct that the setting of the paper is not an operational downscaling setting. There are two reasons for this choice: One is to not add another dataset to the analysis. Apart from the MERRA-2 data we are dealing with ERA5 and CORDEX. In addition, we wanted to avoid confusing raw downscaling performance of the model with questions of bias shift between datasets (unless the high and low resolution data come from the same model run at different resolutions). I.e. the concept is to first show raw downscaling performance and then take this to a more operational setting.
>
> 1e) **Gravity wave parametrization** See separate response.
>
> 1f) **Scaling experiments** See scaling experiments as part of ablation studies in the new appendix.In a nutshell, we obtain about 12% improvement when scaling the model by 8x. Roughly an improvemnt of 4% when doubling the model. Yet please consider the new appendix for details.

---

> ### Author Response · Authors · 2024-11-26
> **Additional remarks regarding gravity wave parametrization**
>
> Most climate model parameterizations, for instance, of clouds, precipitation, etc., are indeed single column, in order to adhere to the single column discretization of the underlying climate models. These processes evolve exclusively within the troposphere (surface-10 km height) at 10-50 km spatial scales and so a single-column approximation works reasonably well for them. Although, recent studies have questioned the accuracy of these simplifications (Wang et al. 2023). Gravity waves on the other hand are excited near surface but propagate much deeper in the stratosphere, and even the mesosphere. At these heights, their effects become prominent, providing first order forcing of the winds. As these waves propagate vertically, they also propagate horizontally (Doppler shifting, refraction, large group velocities) to the extent that by the time they reach the upper troposphere and middle stratosphere, they can propagate upto two to three thousand kilometers in the horizontal (Sato et al. 2012). This significant horizontal propagation been well-established in the scientific literature using a combination of satellite observations, linear-wave theory, and high-resolution models. Hence, there is an active push in the gravity wave research community to develop nonlocal parameterization (Amemiya and Sato 2016, Eichinger et al. 2023, Voelker et al. 2023, Gupta et al. 2024).
>
>  From an AI perspective, since coupling a fast AI emulator in climate models can improve their efficiency, replacing the single column (traditional) scheme with a global ML scheme has the potential to both (a) massively improve the physical representation of gravity waves in climate models, and (b) not compromise on their speed. Recent studies have also explored the middle ground, i.e., having a regionally nonlocal scheme comprising of ANNs which incorporate a nonlocal information stencil to make predictions (Wang et al. 2023).
>
> Thus, our adopted approach is grounded in scientific evidence and the established physics of gravity waves, and based on an increasing body of literature over the past decade, we are confident that having a global scheme or a slightly 'nonlocal' scheme for gravity waves is a promising avenue to proceed.
>
>  References:
>
>  [1] Eichinger, R., Rhode, S., Garny, H., Preusse, P., Pisoft, P., Kuchař, A., Jöckel, P., Kerkweg, A., and Kern, B.: Emulating lateral gravity wave propagation in a global chemistry–climate model (EMAC v2.55.2) through horizontal flux redistribution, Geosci. Model Dev., 16, 5561–5583, https://doi.org/10.5194/gmd-16-5561-2023, 2023.
>
>  [2] Sato, K., S. Tateno, S. Watanabe, and Y. Kawatani, 2012: Gravity Wave Characteristics in the Southern Hemisphere Revealed by a High-Resolution Middle-Atmosphere General Circulation Model. J. Atmos. Sci., 69, 1378–1396, https://doi.org/10.1175/JAS-D-11-0101.1.
>
>  [3] Amemiya, A. and Sato, K., 2016. A new gravity wave parameterization including three-dimensional propagation. Journal of the Meteorological Society of Japan. Ser. II, 94(3), pp.237-256.
>
>  [4] Wang, P., Yuval, J., & O’Gorman, P. A. (2022). Non-local parameterization of atmospheric subgrid processes with neural networks. Journal of Advances in Modeling Earth Systems,14, e2022MS002984. https://doi.org/10.1029/2022MS002984
>
>  [5] Gupta, A., Sheshadri, A., Alexander, M. J., & Birner, T. (2024). Insights on lateral gravity wave propagation in the extratropical stratosphere from 44 years of ERA5 data. Geophysical Research Letters, 51, e2024GL108541. https://doi.org/10.1029/2024GL108541
>
>  [6] Voelker, G. S., Bol¨oni, G., Kim, Y.-H., Z ¨angl, G., and ¨Achatz, U. MS-GWaM: A 3-dimensional transient gravity wave parametrization for atmospheric models, September 2023.

---

> ### Author Response · Authors · 2024-11-26
> **Additional comments regarding minor issues**
>
> Many thanks for your careful remarks concerning minor weaknesses. Our updated version addresses these as follows:
>
> Can you properly discuss, and include a reference to, what a Swin-shift is?
> - It's the defining architectural element of the Swin transformer. But we have added a reference.
>
> Similarly, for the "pixel shuffle layers"
> - See https://pytorch.org/docs/stable/generated/torch.nn.PixelShuffle.html. We don't think it is common to give a reference here.
>
> Line 39: Pangu -> Pangu-Weather
> - Done.
>
> Line 48: Nowcasting should be lower-case
> - Done.
>
> Equation 1: Consider reformulating this as an objective/loss function.
> - The loss would be $RMSE(\hat{X}_{t+\delta t}, X_{t+\delta t})$ with equation (1) solved for $\hat{X}_{t+\delta t}$. If you rewrite it that way, the normalization is harder to understand. I.e. in the current form all quantities appear with the relevant normalization factor $\sigma$ and $\sigma_C$ directly -- the equation is effectively $y / \sigma_Y = f[(x-\mu_x)/sigma_X]$. In other words, if you rewrite this as a loss (as done in GraphCast), it might look easier to implement but harder to interpret.
>
> Also Eq. 1: What is $\hat{X}_t$? What is $\sigma_C$?
> - Done. Added explanation of \hat{X}_t.
> - $\sigma_C$ is explained in the paragraph below.
>
> Line 93: $\sigma_C^2 = \sigma_C^2(X_t-C_T)$ doesn't make sense to me.
> - As it says in the text, the variance of the historical anomaly. I.e. for each variable (+ level), you compute the difference of reanalysis $X_t$ and climate $C_t$. And then you compute the variance across time and space. So while $\sigma$ is computed from $X_t$, $\sigma_C$ is computed from $X_t-C_t$.
>
> Line 104: " same 20 year period that we used for pretraining." .... Do you mean 40 year period? If not, which 20-year period from the 40-year training period did you use?
> - 2000-2019 for the climatology. Edited.
>
> Line 157: Multiple symbols are undefined (e.g. $V_S$).
> - Added explanation.
>
> Line 169: It's not entirely clear what "alternates" means in this context.
> - Figure 1 shows two masking patterns. A global and local one. They are alternated batch to batch during training.
>
> Line 429: "baseline"... do you mean Prithvi WxC?
> - Correct. Edited accordingly.
>
> Line 507: "improved"... improved compared to what?
> - Edited to "strong".
>
> Figure 12: Do you mean 'downscale' on the right "upscale" block?
> - We actually mean "increase in resolution" when writing "upscale" here. Added a clarifying remark to the caption. So this is not a U-net as notably there is some resolution increase before and after "hitting" the pretrained components.
>
> Sections D. 2.3 and D.2.4 in the appendix are literal copies of the corresponding paragraphs on pages 8 and 9. Please remove.
> - Done.

---

> > ### Comment · Reviewer_pzCZ · 2024-11-27
> >
> > Thank you for the response. For future revisions, it would be nice if you could 1) Summarize the new results, and 2) Color-code new results/text in the paper so that it's easier for the reviewer to find the new content (or at least pointers to the line numbers where you've added new stuff).
> >
> > I appreciate your new ablations but many of the points in my review are completely omitted in your response. You're welcome to disagree with me, but please acknowledge each of my points, especially if they are major, of which I list some below (as well as other thoughts based on your response).
> >
> >
> > 1. I want to really encourage the authors to try to create a clearer story on the foundational nature of Prithvi WxC compared to task-specific models. Your MERRA2-FCN baseline is appreciated but doesn't address these points (both because it's a completely different model than Prithvi and because forecasting is part of your pre-training (i.e. not a "downstream task"))**. To quote myself, here are again some of the most relevant baselines that I can think of (as posted in the main review).
> > These experiments would be key to underscore the necessity of a foundation model. Feel free to discuss them with me.
> >
> > > Some highly relevant and simple baselines are:
> > > - How much worse(?) does Prithvi WxC perform on these tasks if you omit the pre-training stage (i.e. initialize with random weights instead of the frozen pre-trained ones, and train all parameters jointly from scratch on the tasks)?
> > > - How about completely removing the pre-trained transformer backbone (i.e. removing the Prithvi WxC block from Figures 12 & 13)?
> > > - For the latter, it would be also good to run an experiment where you replace the pre-trained Prithvi WxC backbone with some "lightweight" blocks (e.g. a (deeper) U-Net), trained in a task-specific way from scratch, to account for the huge difference in parameter counts if you completely remove Prithvi WxC.
> >
> > > These ablations would immensely help in understanding how useful the pre-training stage is for these downstream applications (e.g. does using pre-trained Prithvi WxC improve performance over such simple baselines? Is it more data-efficient?). Besides, otherwise, it is hard to see evidence for the claim in the conclusion that *"Instead of building task-specific ML-models from scratch, these pretrained encoders can be used to develop more precise data-driven models of atmospheric processes"*.
> >
> > **1b: That being said, why is MERRA2-FCN performing so badly? While ERA5-FCN is already not a state-of-the-art model, I'm not sure what would make it perform so badly just by being trained on MERRA2. Is the training recipe any different?
> >
> > 2. Could you expand on what's different, if anything (besides Eq. 1 left side), between your absolute and anomalies ablation? What exactly do you mean by absolute: Is it to predict $X_{t+\delta t}$? Why not standardize it with per-channel (non-temporal) scalars $C, \sigma$, which you do for the inputs and is commonly used in other papers? Also, not all fields seem to improve with climatology which the last column is wrongly implying.
> >
> > 3. Table 7 is nice but why would rollout fine-tuning (i.e. a multi-step loss) improve one-step prediction errors but deteriorate multi-step errors? That's very counter-intuitive. Also, what do these values stand for? Is it some RMSE or loss? If so, which? How and why do you combine 6h, 12h, and 24h instead of having one column for each? Lastly, can you please include Fig. 4 with a line for Prithvi WxC before forecasting fine-tuning? This should be easy since you've already pre-trained that model.
> >
> > 4. Can you respond to my other points about forecasting please?
> >
> > 5. My concerns 1e) 1, 2, 3, and 5 aren't addressed. Thank you though for carefully responding to 1e)-4 on why a global context might be reasonable to have. That would be good to discuss in the paper too.
> >
> > 6. Thanks for including the scaling experiment. That is good to see. I have a minor question on how the loss values translate to downstream performances. That is, should we expect that 12% lower loss will lead to 12% better reconstruction and forecasting performance?
> >
> > 7. My concerns about related work (point 2)) are unaddressed. Similarly, it would be nice to see how the authors plan to discuss the limitations of their work in the revised draft (point (3)). I'm also missing responses to the Questions-Major section of my original review.
> >
> > 8. Thanks for the edits regarding my minor concerns/questions. Two minor follow-ups: a) The reasons line 93 doesn't make sense to me is that $\sigma_C^2$ appears on both sides of the equations but unless $(X_t - C_T)$ is 1, how can that be? b) Please replace upscale with either upsample or downscale. These are the correct terms if you mean to say that you're increasing the resolution.
> >
> > 9. By the way, could you respond to the other reviewers too? I'd be curious to read those discussions.

---

### Author Response · Authors · 2024-11-26
**New version -- summary of changes**

We uploaded a new version of the paper. Changes are as follows
- New appendix (B.2) including **ablation studies** on the use of climatology, the impact of the masking strategy, handling of the lead time signal, the use of context tokens and scaling performance.
- New **baseline for forecasting** results: A version of FourCastNet trained on MERRA-2 data. (Figures 4 and 11.)
- Typos, clarifications and references added in response to referee pzCZ.

---

### Meta-Review · Area_Chair_GbYV · 2024-12-17

**Metareview:**

The paper proposes a foundation model for weather and climate. While the foundation model backbone is similar to many other works in the realm of foundation models, the paper proposes interesting objectives based on the weather domain. The biggest weakness of the paper is in experiments that one of the reviewers discusses at length.

**Additional Comments On Reviewer Discussion:**

My recommendation is based upon high quality discussion amongst the authors and the reviewers. 3 out 4 reviewers considered the paper below the acceptance threshold. A couple of reviewers engaged in a high quality discussions, where one of the reviewers highlighted several ways in which the paper could have been improved. This included ablation studies and more experimentation. The authors did address some these concerns in their rebuttal but the reviewer felt that not all of his points were addressed. The most positive reviewer did not engage in the conversation and did not champion for the paper.

---

### Decision · Program_Chairs · 2025-01-22

Reject